# Divergence-Regularized Discounted Aggregation: Equilibrium Finding in Multiplayer Partially Observable Stochastic Games

**Runyu Lu**[1,2]**, Yuanheng Zhu**[2,1]**, Dongbin Zhao**[2,1]
[1] School of Artificial Intelligence, University of Chinese Academy of Sciences
[2] Institute of Automation, Chinese Academy of Sciences
`lurunyu17@mails.ucas.ac.cn`
`{yuanheng.zhu,dongbin.zhao}@ia.ac.cn`

## ABSTRACT

This paper presents Divergence-Regularized Discounted Aggregation (DRDA), a multi-round learning system for solving partially observable stochastic games (POSGs). DRDA is based on action values and applicable to multiplayer POSGs, which can unify normal-form games (NFGs), extensive-form games (EFGs) with perfect recall, and Markov games (MGs). In each single round, DRDA can be viewed as a discounted variant of Follow the Regularized Leader (FTRL) under a general value function for POSGs. While previous studies on discounted FTRL have demonstrated its last-iterate convergence towards quantal response equilibrium (QRE) in NFGs, this paper extends the theoretical results to POSGs under divergence regularization and generalizes the QRE concept of Nash distribution. The linear last-iterate convergence of single-round DRDA to its rest point is proved under the assumption on the hypomonotonicity of the game. When the rest point is unique, it induces the unique Nash distribution defined in the POSG, which has a bounded deviation from Nash equilibrium (NE). Under multiple learning rounds, DRDA keeps replacing the base policy for divergence regularization with the policy at the rest point in the previous round. It is further proved that the limit point of multi-round DRDA must be an exact NE (rather than a QRE). In experiments, discrete-time DRDA can converge to NE at a near-exponential rate in (multiplayer) NFGs and outperform the existing baselines for EFGs, MGs, and typical POSGs.

## 1 INTRODUCTION

While a wide range of game-theoretic learning dynamics, including no-regret dynamics and best-response dynamics, are primarily analyzed in static normal-form games (NFGs), many real-world games are dynamic and usually considered under a different game representation. For example, simultaneous games like pursuit-evasion games (Li et al., 2024) and fighting video games (Tang et al., 2023) can be formulated as Markov games (MGs), especially when there is an infinite horizon. Imperfect-information sequential games like Texas Hold'em are commonly formulated as extensive-form games (EFGs) with perfect recall. In view of this requirement, many of the recent studies have tried to extend the theoretical results established in NFGs to dynamic games (Kroer et al., 2020; Farina et al., 2021b). Counterfactual regret minimization (CFR) (Zinkevich et al., 2007), as a well-known example, decomposes global regrets into local ones under counterfactual value and enables regret minimization within a time complexity polynomial in the size of the game tree. However, when it comes to infinite-horizon games like MGs, such generalizations are not feasible. Instead, another line of algorithms based on best-response computations, including Fictitious Play (FP) (Brown, 1951), Policy Space Response Oracle (PSRO) (Lanctot et al., 2017), and Exploitability Descent (ED) (Lockhart et al., 2019), are known to be applicable. As an extension of MG, partially observable stochastic game (POSG) introduces imperfect information and is capable of unifying NFG, (perfect-recall) EFG, and MG. However, it is not clear if a basic learning dynamic under general action value can be directly applied to POSGs without sacrificing its convergence properties in NFGs.

In recent years, last-iterate convergence has become an increasingly interesting research target (Wei et al., 2021; Lee et al., 2021). Compared to the well-examined concept of average-iterate convergence, which CFR and FP are based upon, last-iterate convergence means that there is no need to preserve policies in time average. This is an ideal property for further extension to deep reinforcement learning (DRL), as it is intractable to time-average function approximators like neural networks. Much of the existing research on last-iterate convergence is related to Follow the Regularized Leader (FTRL) (Shalev-Shwartz & Singer, 2006)), which is a basic learning framework that has a close relationship with various learning dynamics in game theory. The widely studied replicator dynamics (Taylor & Jonker, 1978) as well as Hedge, multiplicative weights, and gradient methods can be captured as special cases of FTRL (Cesa-Bianchi & Lugosi, 2006; Arora et al., 2012; Hazan et al., 2016). While it is proved that vanilla FTRL can cycle in NFGs and EFGs (Mertikopoulos et al., 2018; Perolat et al., 2021), some of its variants exhibit last-iterate convergence. For example, Daskalakis & Panageas (2019) prove that the optimistic variant of FTRL exhibits last-iterate convergence to Nash equilibrium (NE) in the two-player zero-sum games with the unique equilibrium. To avoid the uniqueness assumption, some recent works also focus on solving the "regularized" game and directly establishing last-iterate convergence to the quantal response equilibrium (QRE, which can be viewed as an approximation to NE) (McKelvey & Palfrey, 1995) of the original game. These results are applicable to EFGs (Liu et al., 2023; Sokota et al., 2023) or MGs (Cen et al., 2021; 2023), but they are generally restricted to two-player zero-sum games. To deal with multiplayer games, we also focus on some "discounted" variants besides the works on reward regularization.

A discounted variant of FTRL, first examined in Leslie & Collins (2005), is proved to approximate Nash equilibrium under a shrinking temperature parameter in two-player zero-sum MGs (Sayin et al., 2021). Even for multiplayer games, existing theory suggests that it can converge to the solution concept of Nash distribution, a specific form of QRE defined in NFGs, under a range of distinct assumptions (Coucheney et al., 2015; Gao & Pavel, 2021). While discounted FTRL exhibits strong convergence properties, it is still at the price of applying a "perturbation" to the equilibrium point. Actually, it is completely unknown whether discounted FTRL can approximate exact Nash equilibrium in multiplayer POSGs. Recently, a series of works on two-player zero-sum or monotone games (Perolat et al., 2021; Abe et al., 2022; 2023) have shown that the perturbation arising from divergence-regularized rewards can be mitigated using a technique of multi-round learning, which repeatedly replaces the base policy (or "anchor" policy; see Jacob et al. (2022); Bakhtin et al. (2023)) with the policy at the rest point in each round. Inspired by these results, we introduce a multi-round equilibrium-learning system named Divergence-Regularized Discounted Aggregation (DRDA), based on the continuous-time dynamic of discounted FTRL. We further examine the last-iterate convergence of DRDA in the POSGs with certain hypomonotonicity (a property more extensively examined in optimization; see Alomar & Chavdarova (2024)), as well as its relationship to Nash equilibrium.

Specifically, the contribution of this paper includes:

- Employing the advantage value (a type of action value) at each decision point in POSGs, we propose single-round DRDA as an extension of (NFG-based or MG-based) discounted FTRL. By defining a generalized Nash distribution (GND) that extends Nash distribution to POSGs, we first prove the relationship between the rest point of a learning dynamic and a QRE concept in POSGs: every GND induces a DRDA rest point in POSGs (Theorem 1).

- By assuming local hypomonotonicity, which is a relaxation of the widely used concept of global monotonicity (Gorbunov et al., 2022b; Cai et al., 2022), we prove that single-round DRDA converges to its rest point in any local $\lambda$-hypomonotone region at a linear rate when the regularization (temperature) parameter $\epsilon$ in DRDA is greater than $\lambda$ (Theorem 2). Compared to the existing works (Perolat et al., 2021; Sokota et al., 2023; Abe et al., 2024), our convergence result is directly based on the action value for POSGs (possibly with imperfect information and an infinite horizon), while requiring a less stringent assumption.

- By assuming the uniqueness of DRDA rest point, we establish an explicit NashConv bound $\epsilon n \log \mathcal{K} \sum_t \gamma^t$ for the rest point of single-round DRDA (Theorem 3). By showing that this gap can vanish as the rest-point sequence of DRDA converges in multiple rounds, we further prove that the limit point of multi-round DRDA is an exact Nash equilibrium (Theorem 4).

- Through experiments, we show that a discrete-time implementation of multi-round DRDA converges to Nash equilibria in various finite-action games under the POSG formulation. For NFGs, DRDA can converge at a near-exponential rate. For multiplayer EFGs, DRDA can

outperform the baselines including extragradient CFR$^+$ (Farina et al., 2024), moving-magnet magnetic mirror decent (Sokota et al., 2023), and regularized Nash dynamics (Perolat et al., 2022). Besides, DRDA significantly outperforms the general methods in an infinite-horizon Markov game and the existing baselines in typical POSG environments.

## 2 PRELIMINARIES

### 2.1 PARTIALLY OBSERVABLE STOCHASTIC GAME

We use the formulation of partially observable stochastic games (POSGs) as a general framework to describe a class of finite-action games possibly with imperfect information and an infinite horizon.

**Definition 1.** *A **partially observable stochastic game** is a tuple $\left\langle N, S, O, \mathcal{A}, \mathcal{P}, \mathcal{Z}, \left\{ R^i \right\}, \rho, \gamma, T \right\rangle$:*

$N = \{1, 2, \cdots, n\}$ *is the set of all players. $S$ is the set of all global states $s \in S$. $O$ is the set of all joint observations $\vec{o} = (o^i)_{i \in N} \in O$. $\mathcal{A}$ is the set of all players' actions, with the joint action expressed as $\vec{a} = (a^i)_{i \in N} \in \mathcal{A}^n$. $\mathcal{P}$ is the state transition distributions, with a subsequent state generated by $s_{t+1} \sim \mathcal{P}(s_t, \vec{a}_t)$. $\mathcal{Z}$ is the observation distributions, with a joint observation generated by $\vec{o}_{t+1} \sim \mathcal{Z}(s_t, \vec{a}_t, s_{t+1})$. $R^i$ is the reward function for player $i$, with an instant reward generated by $r_{t+1}^i = R^i(s_t, \vec{a}_t, s_{t+1})$. $\rho$ is the initial state distribution, assigning each $s \in S$ a probability $\rho(s)$ to be the initial state $s_0$. $\gamma \in (0, 1]$ is the discount factor, and $T$ is the termination time. For finite-horizon games, $\gamma = 1$, and $T$ is finite. For infinite-horizon games, $\gamma < 1$, and $T \to \infty$.*

Inherently, POSG is capable of describing simultaneous games like NFGs and MGs. Furthermore, since sequential moves can be regarded as simultaneous moves with empty actions for the waiting players, sequential games like EFGs can also be converted to equivalent POSGs.

### 2.2 BASIC CONCEPTS & NOTATIONS

The following concepts and notations will be frequently used in this paper.

**History.** As POSG contains imperfect information, the true state of the game at time step $t \leq T$ should be expressed as a *history* $h_t = (s_0, \vec{o}_0, \vec{a}_0, s_1, \vec{o}_1, \vec{a}_1, \cdots, s_t, \vec{o}_t) \in \mathcal{H}$ rather than a single current state $s_t$. As a comparable concept, the *trajectory* used in the field of reinforcement learning (RL) can be viewed as a history with rewards: $\tau_t = (s_0, \vec{o}_0, \vec{a}_0, s_1, \vec{r}_1, \vec{o}_1, \vec{a}_1, \cdots, s_t, \vec{r}_t, \vec{o}_t)$.

**Decision point.** In a POSG, player $i$ makes his/her decisions based on his/her past observations and actions $x_t^i = \left(o_0^i, a_0^i, o_1^i, a_1^i, \cdots, o_t^i\right)$. In the case of imperfect-information games, each $x_t^i$ can be viewed as an *information set* and corresponds to multiple true histories $h_t$. In the case of infinite-horizon games, some $x_t^i$ (even with different $t$) are equivalent under the optimal strategy and can be regarded as the same *decision point*, which we use any one of the $x_t^i$ (or $x^i$) to denote. For example, in a Markov game, player $i$ is at the same decision point when the current observation (the last term in $x^i$) is the same. We use $\mathcal{X}^i$ to denote the set of all decision points and require the number of distinct decision points $|\mathcal{X}^i|$ to be finite throughout this paper.

We borrow the notation from the literature of EFGs and write $h_t \in x_t^i$ (viewing $x_t^i$ as an information set) if $h_t$ is not contradictory to the existing terms in $x_t^i$. For $k \geq t$, we write $h_k \sqsupseteq h_t$ to express $h_k$ as a successor of $h_t$ in the game tree. Similarly, we write $x_k^i \sqsupseteq x_t^i$ to express $x_k^i$ as a successor of $x_t^i$.

In the special case of perfect-information games, the $x_t^i$ of any player $i$ can uniquely determine $h_t$. Furthermore, we can simply use $s_t$ to represent both $h_t$ and $x_t^i$ in this case, as the POSG is reduced to an MG. The expression of NFG, EFG, and MG under the POSG formulation is shown in Table 1.

Table 1: Conversion of NFG, EFG, and MG under the framework of POSG

|  | NFG | EFG | MG |
|---|---|---|---|
| Termination Time | $T = 1$ | $T = $ depth of game tree | $T \to \infty$ |
| Discount Factor | $-$ | $\gamma = 1$ | $\gamma < 1$ |
| Dynamics Reduction | $\|S\| = \|O\| = 1$ | $\mathcal{P}(s_t, \vec{a}_t) = \mathcal{P}(s_t, a_t^{i(s_t)})$ | $x_t^i = h_t = s_t$ |

● For EFGs, we use $i(s_t)$ to denote the current player under global state $s_t$.

**Policy.** At each decision point $x^i$, player $i$'s *policy* $\pi^i(x^i)$ is a probability distribution over $\mathcal{A}$. Player $i$'s complete policy is a combination of the policies at all $x^i \in \mathcal{X}^i$. We use the joint policy $\vec{\pi} = (\pi^i)_{i \in N} \in \Pi$ to denote the combination of all players' complete policies and use $\vec{\pi}^{-i}$ to denote the combination of all players' policies except player $i$'s. By Definition 1, the probability of visiting history $h_t = (s_0, \vec{o}_0, \vec{a}_0, s_1, \vec{o}_1, \vec{a}_1, \cdots, s_t, \vec{o}_t)$ under a joint policy $\vec{\pi}$ is computed as follows:

$$\Pr(h_t | \vec{\pi}) = \rho(s_0) \mathcal{Z}(s_0, \vec{o}_0) \sum_{k=0}^{t-1} \left( \prod_{i=1}^{n} \pi^i(x_k^i, a_k^i) \right) \mathcal{P}((s_k, \vec{a}_k), s_{k+1}) \mathcal{Z}((s_k, \vec{a}_k, s_{k+1}), \vec{o}_{k+1})$$

**Value functions & Advantage.** In POSGs, the *value function* of history $h_t$ under a joint policy $\vec{\pi}$ is defined as $V_{\vec{\pi}}^i(h_t) = \mathbb{E}\left[ \sum_{k=t}^{T-1} \gamma^{k-t} r_{k+1}^i | h_t, \vec{\pi} \right]$. The corresponding *state-action value function* is defined as $Q_{\vec{\pi}}^i(h_t, a_t^i) = \mathbb{E}\left[ r_{t+1}^i + \gamma V_{\vec{\pi}}^i(h_{t+1}) | h_t, a_t^i, \vec{\pi} \right]$. Borrowing the notation from RL literature (Sutton & Barto, 2018), define history-based *advantage* as $A_{\vec{\pi}}^i(h_t, a_t^i) = Q_{\vec{\pi}}^i(h_t, a_t^i) - V_{\vec{\pi}}^i(h_t)$.

**Utility & Nash equilibrium.** The individual *utility* in POSGs is defined as the expected value over the initial states: $u^i(\vec{\pi}) = \mathbb{E}_{h_0}\left[ V_{\vec{\pi}}^i(h_0) \right]$, where $h_0 = (s_0, \vec{o}_0)$, with $s_0 \sim \rho$ and $\vec{o} \sim \mathcal{Z}(s_0)$.

As a commonly used solution concept in game theory, *Nash equilibrium* (NE) is a joint policy $\vec{\pi}_{nash}$ where no player can increase his/her own utility by unilaterally deviating from his/her own policy. Specifically, for any player $i \in N$ and any individual policy $\pi^i \in \Pi^i$, it holds that $u^i(\vec{\pi}_{nash}) \geq u^i(\pi^i, \vec{\pi}_{nash}^{-i})$. Based on the Nikaido-Isoda function (Nikaidô & Isoda, 1955), define NashConv$(\vec{\pi}) = \sum_{i=1}^{n} \max_{\pi\dagger^i \in \Pi^i} \{ u^i(\pi\dagger^i, \vec{\pi}^{-i}) - u^i(\vec{\pi}) \}$. NashConv measures the deviation of $\vec{\pi}$ with respect to Nash equilibrium. Clearly, NashConv$(\vec{\pi}_{nash}) = 0$.

## 2.3 FOLLOW THE REGULARIZED LEADER & DISCOUNTED FTRL

**Follow the Regularized Leader.** Follow the Regularized Leader (FTRL) is a widely examined equilibrium-learning dynamic. In NFGs, the continuous-time FTRL (Mertikopoulos et al., 2018) can be expressed as an ODE of the score $\vec{y}$ with a non-negative continuous time variable $t$ (different from the discrete time step $t$ used in the formulation of POSG):

$$\begin{cases} \dot{y}_t^i = w^i(\vec{\pi}_t) \\ \pi_t^i = \sigma^i(y_t^i) \end{cases} \tag{1}$$

where $w^i(\vec{\pi}_t)(a) = u^i(\pi_a^i, \vec{\pi}_t^{-i})$ is player $i$'s expected utility under a pure strategy $\pi_a^i(\cdot)$ with $\pi_a^i(a) = 1$, and $\sigma^i(y_t^i) = \arg\max_{\pi^i \in \Delta(\mathcal{A})} \{ \langle \pi^i, y_t^i \rangle - \phi^i(\pi^i) \}$ is the policy selection that maps the current score function $y_t^i(\cdot)$ into the policy space $\Delta(\mathcal{A})$.

The penalty function $\phi^i(\pi^i)$ guarantees that the $\sigma^i(\cdot)$ in (1) is well-defined in the sense that the *argmax* results in a singleton. Under entropic regularization (i.e., $\phi^i(\pi^i) = \epsilon \sum_{a^i \in \mathcal{A}} \pi^i(a^i) \log \pi^i(a^i)$), FTRL is equivalent to multiplicative weights (Hedge) (Cesa-Bianchi & Lugosi, 2006; Arora et al., 2012) as well as replicator dynamics in evolutionary game theory (Taylor & Jonker, 1978).

**Discounted FTRL.** While FTRL exhibits average-iterate convergence, the last iterate of FTRL may not converge. Actually, it features a cycling behavior that is robust to the choice of regularization, utility transformations, and game restrictions (see Mertikopoulos et al. (2018)). To avoid the recurrence behavior, a "discounted" variant of FTRL can be used instead (Leslie & Collins, 2005):

$$\begin{cases} \dot{y}_t^i = w^i(\vec{\pi}_t) - y_t^i \\ \pi_t^i = \sigma^i(y_t^i) \end{cases} \tag{2}$$

Discounted FTRL uses $w^i(\vec{\pi}_t) - y_t^i$ instead of $w^i(\vec{\pi}_t)$ as the derivative of the score $\vec{y}_t$. Consequently, the past aggregation of utility will be discounted as $t$ increases. This guarantees last-iterate convergence towards the solution concept of *Nash distribution* in NFGs (Gao & Pavel, 2021; Coucheney et al., 2015). Nash distribution is defined as the NE policy under a utility function perturbed by entropic regularization $\tilde{u}^i(\vec{\pi}) = u^i(\vec{\pi}) - \phi^i(\pi^i)$, where $\phi^i(\pi^i) = \epsilon \sum_{a^i \in \mathcal{A}} \pi^i(a^i) \log \pi^i(a^i)$. It is a specific form of quantal response equilibrium (QRE) (McKelvey & Palfrey, 1995) and can be viewed as a relaxation of the original NE concept.

**Rest point.** To formally examine the last-iterate convergence of a continuous-time dynamic, we need to use the concept of *rest point*. For discounted FTRL (2), $(\vec{y}_r, \vec{\pi}_r)$ is a rest point if both $y_r^i = w^i(\vec{\pi}_r)$ and $\pi_r^i \in \sigma^i(y_r^i)$ are satisfied for all $i \in N$. Intuitively, it means that $\vec{y}_t \equiv \vec{y}_r, \vec{\pi}_t \equiv \vec{\pi}_r$ $(t \geq 0)$ is a solution to the ODE. Under entropic regularization, it can be proved that the policy at the rest point of discounted FTRL is equivalent to the concept of Nash distribution in NFGs (see Gao & Pavel (2021)).

# 3 DIVERGENCE-REGULARIZED DISCOUNTED AGGREGATION

As we have mentioned, while the last-iterate convergence property of discounted FTRL is desirable, it is restricted to NFGs and comes at the price of perturbing the equilibrium point. In view of this, we further employ a general action value and combine the idea of multi-round learning to construct Divergence-Regularized Discounted Aggregation (DRDA), a learning system for solving POSGs.

## 3.1 SINGLE-ROUND DRDA

As POSG concerns imperfect information, define *advantage value* $v^i(\vec{\pi})$ as a kind of action value that averages all histories' advantages at a decision point based on the visitation probability $\Pr(h|\vec{\pi})$:

$$v^i(\vec{\pi})(x^i, a^i) = \frac{\sum\limits_{h \in x^i} \Pr(h|\vec{\pi}) A_{\vec{\pi}}^i(h, a^i)}{\sum\limits_{h \in x^i} \Pr(h|\vec{\pi})} \tag{3}$$

With the definition of $v^i(\vec{\pi})$, we analyze the following decision-point-level ODE in a POSG.

**Definition 2.** *A **single-round DRDA** under regularization parameter $\epsilon > 0$ and base policy $\vec{\pi}_{base}$ is a continuous-time dynamic expressed as the following ODE with a non-negative time variable $t$:*

$$\begin{cases} \dot{y}_t^i = v^i(\vec{\pi}_t) - y_t^i \\ \pi_t^i = \sigma^i(y_t^i) \end{cases} \tag{4}$$

*where $y_t^i(x^i, \cdot)$ is the score function at the decision point $x^i \in \mathcal{X}^i$, $v^i(\vec{\pi}_t)(x^i, \cdot)$ is the advantage value defined by (3), and $\sigma^i(y_t^i)(x^i)$ is a policy choice map under divergence regularization:*

$$\sigma^i(y_t^i)(x^i) = \underset{\pi^i(x^i) \in \Delta(\mathcal{A})}{\arg\max} \left\{ \langle \pi^i(x^i), y_t^i(x^i) \rangle - \epsilon D_{\mathrm{KL}}\left(\pi^i(x^i) || \pi_{base}^i(x^i)\right) \right\} \tag{5}$$

Note that the advantage value $v^i(\vec{\pi}_t)$ in the single-round DRDA (4) is comparable to $w^i(\vec{\pi}_t)$ in the discounted FTRL (2). When $T = |S| = |O| = 1$, the POSG is reduced to an NFG, and we have $v^i(\vec{\pi}_t)(x_0^i, a) = w^i(\vec{\pi}_t)(a) - u^i(\vec{\pi}_t)$. From the integral form of the score function, we can see that a single-round DRDA is a discounted aggregation of the past $v^i(\vec{\pi}_t)$ under a factor of $e^{-(t-\tau)}$:

$$y_t^i(x^i, a^i) = e^{-t} y_0^i(x^i, a^i) + \int_0^t e^{-(t-\tau)} v^i(\vec{\pi}_\tau)(x^i, a^i) d\tau \tag{6}$$

Different from vanilla FTRL, the introduction of the exponentially decaying factor suggests the boundedness of the score function:

**Lemma 1.** *In single-round DRDA, the score $\vec{y}$ is bounded.*

Besides, we use KL-divergence $D_{\mathrm{KL}}\left(\pi^i(x^i) || \pi_{base}^i(x^i)\right) = \sum_{a^i \in \mathcal{A}} \pi^i(x^i, a^i) \log \frac{\pi^i(x^i, a^i)}{\pi_{base}^i(x^i, a^i)}$ as the regularizer (penalty) function in (5) and requires $\vec{\pi}_{base}$ to be of full support. Correspondingly, we have an equivalent expression for $\sigma^i(y_t^i)$, which is in a form similar to the well-known *softmax*:

**Lemma 2.** *In single-round DRDA, $\sigma^i(y_t^i)$ has a closed-form expression:*

$$\sigma^i(y_t^i)(x^i, a^i) = \frac{\pi_{base}^i(x^i, a^i) \exp\left(\frac{1}{\epsilon} y_t^i(x^i, a^i)\right)}{\sum\limits_{b \in \mathcal{A}} \pi_{base}^i(x^i, b) \exp\left(\frac{1}{\epsilon} y_t^i(x^i, b)\right)} \tag{7}$$

It is direct to verify that the advantage value $\vec{v}(\cdot)$ is continuous. With the continuity of $\vec{\sigma}(\cdot)$ (7), we know that $\vec{v}(\vec{\sigma}(\cdot))$ is also continuous. By Brouwer's fixed-point theorem (Florenzano, 2003), a fixed point of $\vec{v}(\vec{\sigma}(\cdot))$ exists. The existence of the rest point of single-round DRDA is directly proved.

## 3.2 MULTI-ROUND DRDA & DISCOUNTED AGGREGATION

Based on Definition 2, we further construct multi-round DRDA to extend the learning process to multiple rounds, using an idea of repeatedly replacing the base policy $\vec{\pi}_{base}$ (see Perolat et al. (2021); Abe et al. (2022; 2023)). This technique can help the overall learning process to approach Nash equilibrium rather than QRE. Specifically, a *round* corresponds to a time-evolving process of (4) until reaching its rest point (equilibrium point). The ODEs in different rounds differ in the base policy $\vec{\pi}_{base}$ for divergence regularization.

**Definition 3.** *A **multi-round DRDA** under regularization parameter $\epsilon > 0$ and initial point $p_0$ is an iterative process of calling an oracle $\mathcal{M}$. In the $l$-th round ($l \geq 1$), $\mathcal{M}$ takes $(\vec{y}_0, \vec{\pi}_0) = p_{l-1}$ as an input and outputs $p_l = (\vec{y}_r, \vec{\pi}_r)$, where $(\vec{y}_r, \vec{\pi}_r)$ is the rest point of the single-round DRDA starting from $(\vec{y}_0, \vec{\pi}_0)$ under the specified $\epsilon$ and $\vec{\pi}_{base} = \vec{\pi}_0$.*

To examine if multi-round DRDA can find the exact Nash equilibrium rather than QRE, we also need to define the counterpart of single-round DRDA under $\epsilon = 0$. We call it Discounted Aggregation (DA), which corresponds to the problem of $\dot{y}_t^i = v^i(\vec{\pi}_t) - y_t^i$ under the restriction:

$$\pi_t^i(x^i) \in \underset{\pi^i(x^i) \in \Delta(\mathcal{A})}{\arg\max} \left\{ \left\langle \pi^i(x^i), y_t^i(x^i) \right\rangle \right\}, \forall x^i \in \mathcal{X}^i \tag{8}$$

Note that the *hardmax* policy selection above implies that $\pi_t^i$ is a satisfactory policy when only the actions with the highest score $y_t^i$ are assigned non-zero probability. In Section 4, we will see that every NE in the POSG actually corresponds to a solution of DA. Furthermore, when the solution (rest point) of DA is unique, the POSG has a unique NE corresponding to the unique rest point.

## 4 THEORETICAL ANALYSIS

In this section, we examine the convergence properties of DRDA. We will characterize the rest point of single-round DRDA with a QRE concept in POSGs and prove a hypomonotinicity-based condition for single-round DRDA to converge to the rest point at a linear rate. Furthermore, we will show that under the uniqueness assumption on DA rest point, the limit point of multi-round DRDA is actually an exact Nash equilibrium rather than a QRE. All omitted proofs are provided in Appendix C.

### 4.1 GENERALIZED NASH DISTRIBUTION

As a type of quantal response equilibrium (QRE) (McKelvey & Palfrey, 1995), Nash distribution (Leslie & Collins, 2005) in NFGs corresponds to a solution concept close to Nash equilibrium but under certain utility perturbation. Here, we first generalize this concept in the context of POSGs.

**Definition 4.** *A **generalized Nash distribution** (GND) under regularization parameter $\epsilon \geq 0$ and base policy $\vec{\pi}_{base}$ is a joint policy $\vec{\pi}_*$. For any player $i \in N$ and any individual policy $\pi^i \in \Pi^i$:*

$$u^i(\vec{\pi}_*) - \epsilon \sum_{t=0}^{T-1} \gamma^t \sum_{x_t^i} \left( D_{\mathrm{KL}} \left( \pi_*^i(x_t^i) || \pi_{base}^i(x_t^i) \right) \sum_{h_t \in x_t^i} \Pr\left( h_t | \vec{\pi}_* \right) \right) \geq$$

$$u^i(\pi^i, \vec{\pi}_*^{-i}) - \epsilon \sum_{t=0}^{T-1} \gamma^t \sum_{x_t^i} \left( D_{\mathrm{KL}} \left( \pi^i(x_t^i) || \pi_{base}^i(x_t^i) \right) \sum_{h_t \in x_t^i} \Pr\left( h_t | \vec{\pi}_* \right) \right)$$

Note that the GND introduces a utility perturbation based on the KL-divergence under $\vec{\pi}_{base}$ and the visitation probability $\Pr\left( h_t | \vec{\pi}_* \right)$ under $\vec{\pi}_*$. When $\epsilon = 0$, the GND becomes a Nash equilibrium. Also note that when $T = |S| = |O| = 1$, the POSG is reduced to an NFG, and the GND is equivalent to the concept of Nash distribution defined in NFGs if $\vec{\pi}_{base}$ is set to be a uniform policy. However, please note that the GND defined in POSGs is *not* the same as the Nash distribution defined in the (exponentially large) NFG representation of the POSG.

Now we use the following lemma to show an important inequality for GND. This inequality connects the solution concept of GND with the rest point of single-round DRDA. Its proof involves applying some existing results about action values to POSGs (see Appendices B.1 and C.3).

**Lemma 3.** *Given a GND $\vec{\pi}_*$ under $\epsilon$ and $\vec{\pi}_{base}$, it holds for any player $i \in N$, any individual policy $\pi^i \in \Pi^i$, and any individual decision point $x_t^i \in \mathcal{X}^i$:*

$$\sum_{h_t \in x_t^i} \Pr\left(h_t | \vec{\pi}_*\right) \left( \sum_{a_t^i} \pi_*^i(x_t^i, a_t^i) A_{\vec{\pi}_*}^i(h_t, a_t^i) - \epsilon D_{\mathrm{KL}}\left(\pi_*^i(x_t^i) || \pi_{base}^i(x_t^i)\right) \right) \geq$$

$$\sum_{h_t \in x_t^i} \Pr\left(h_t | \vec{\pi}_*\right) \left( \sum_{a_t^i} \pi^i(x_t^i, a_t^i) A_{\vec{\pi}_*}^i(h_t, a_t^i) - \epsilon D_{\mathrm{KL}}\left(\pi^i(x_t^i) || \pi_{base}^i(x_t^i)\right) \right)$$

The inequality above is further used to prove the following theorem, which claims that every GND must correspond to a rest point of our learning dynamic. This theorem will be subsequently used to relate the rest point back to the solution concept of Nash equilibrium.

**Theorem 1.** *In a POSG, every GND $\vec{\pi}_*$ under $\epsilon > 0$ induces a rest point $(\vec{v}(\vec{\pi}_*), \vec{\pi}_*)$ in single-round DRDA, and every GND $\vec{\pi}_*$ under $\epsilon = 0$ (i.e., Nash equilibrium) induces a DA rest point $(\vec{v}(\vec{\pi}_*), \vec{\pi}_*)$.*

## 4.2 CONVERGENCE OF SINGLE-ROUND DRDA

Now we analyze the conditions for single-round DRDA to converge to its rest point in POSGs. Our analysis primarily relies on two properties: the strong convexity of the penalty function and the local hypomonotonicity of the game. First, we need to show that, similar to other forms of regularization used in FTRL, the divergence regularization used in single-round DRDA (5) is 1-strongly convex.

**Definition 5.** *A function $f(\cdot)$ on a compact convex set $C$ is $K$-strongly convex if for any $z, z' \in C$ and any $\beta \in [0, 1]$, $f(\alpha z + (1 - \alpha)z') \leq \alpha f(z) + (1 - \alpha)f(z') - \frac{1}{2}K\alpha(1 - \alpha)\|z - z'\|^2$.*

**Lemma 4.** *$D_{\mathrm{KL}}(\pi || \mu) = \sum_a \pi(a) \log \frac{\pi(a)}{\mu(a)}$ is 1-strongly with respect to $\pi$ under $\ell_1$ norm.*

Second, we need to examine the *hypomonotonicity* of the game. This property is a bit different from cocoercivity, which is also used, e.g., in analyzing the convergence of the extragradient method (Gorbunov et al., 2022a). The concept of hypomonotonicity relaxes *monotonicity*, a more strict but widely used concept in the existing convergence analyses (Gorbunov et al., 2022b; Cai et al., 2022).

**Definition 6.** *An operator $F: \mathbb{R}^d \to \mathbb{R}^d$ is called $\boldsymbol{\lambda}$-hypomonotone ($\lambda \geq 0$) if for any $x, x' \in \mathbb{R}^d$:*

$$\langle x - x', F(x) - F(x') \rangle \geq -\lambda \|x - x'\|_2^2 \tag{9}$$

Note that being 0-hypomonotone means being monotone. With the hypomonotonicity of a proper value operator, we can define the hypomonotonicity of a POSG on a local policy set.

**Definition 7.** *A POSG is locally $\lambda$-hypomonotone ($\lambda \geq 0$) on a policy set $\Pi_{local} \subset \Pi$ if the negative advantage value $-\vec{v}(\cdot)$ is $\lambda$-hypomonotone ($\lambda \geq 0$) on $\Pi_{local}$. I.e., for any $\vec{\pi}_1, \vec{\pi}_2 \in \Pi_{local}$:*

$$\sum_{i \in N} \langle \pi_1^i - \pi_2^i, v^i(\vec{\pi}_1) - v^i(\vec{\pi}_2) \rangle \leq \lambda \sum_{i \in N} \|\pi_1^i - \pi_2^i\|_2^2 \tag{10}$$

*where the inner product sums over all $(k, h_k, a_k^i)$ triples.*

Using the strong convexity of the penalty function, we can now derive a general condition for the linear convergence of single-round DRDA towards its rest point under local hypomonotonicity.

**Theorem 2.** *Assume that a rest point $(\vec{y}_\epsilon, \vec{\pi}_\epsilon)$ of single-round DRDA under $\epsilon > 0$ is contained in a positively invariant compact set $\Omega$ that induces a local policy set $\Pi_{local} = \{\vec{\sigma}(\vec{y}) \,|\, \vec{y} \in \Omega\}$, where $\lambda$-hypomonotonicity is satisfied. If $\epsilon > \lambda$, then the policy of the DRDA starting from an arbitrary $\vec{y}_0 \in \Omega$ will converge to $\vec{\pi}_\epsilon$ at a linear rate. Specifically, there exists a non-negative energy function $\mathcal{V}(\vec{y}_t)$ satisfying $\dot{\mathcal{V}}(\vec{y}_t) \leq -2(1 - \frac{\lambda}{\epsilon})\mathcal{V}(\vec{y}_t)$, with $\vec{\pi} = \vec{\sigma}(\vec{y}) = \vec{\pi}_\epsilon$ when $\mathcal{V}(\vec{y}) = 0$.*

The energy function $\mathcal{V}(\vec{y}_t)$ is constructed based on a concept called Fenchel coupling (Mertikopoulos & Zhou, 2019), which has a lower bound under a strongly convex penalty function (e.g., KL-divergence). We decompose the derivative of the energy function into one term that can be upper-bounded using the assumption of hypomonotonicity and two negative Fenchel coupling terms. Under

$\epsilon > \lambda$, the derivative is non-positive and the energy function approaches zero exponentially fast. We complete the proof by showing that the policy at the points with zero energy is exactly $\vec{\pi}_\epsilon$.

Note that *local hypomonotonicity* only requires the inequality (10) to hold in a local region of policies $\Pi_{local} \subset \Pi$. When $\Pi_{local} = \Pi$, we say the game is globally hypomonotone. Actually, the following proposition guarantees that *global hypomonotonicity* is a common property for at least all NFGs.

**Proposition 1.** *Every NFG is globally $\lambda$-hypomonotone under a sufficiently large $\lambda$.*

To further demonstrate that hypomonotonicity can still be a realistic assumption when the game is dynamic, we also numerically estimate the order of magnitude of the global hypomonotonicity value $\lambda$ by sampling $\frac{\sum_{i \in N} \langle \pi_1^i - \pi_2^i, v^i(\vec{\pi}_1) - v^i(\vec{\pi}_2) \rangle}{\sum_{i \in N} \left\| \pi_1^i - \pi_2^i \right\|_2^2}$. We provide the numerical results in Appendix D.

### 4.3 RELATIONSHIP TO NASH EQUILIBRIUM

Recall that, under $\epsilon = 0$, Theorem 1 implies that every Nash equilibrium induces a rest point of DA (8). Thus, the existence of the DA rest point can be guaranteed by the existence of NE. Furthermore, if the rest point is unique, then the policy at the rest point must be the unique Nash equilibrium of the game. Under this uniqueness assumption, the problem of finding Nash equilibrium can be reduced to finding the rest point of DA. Formally, we have the following theorem:

**Theorem 3.** *If the rest point $(\vec{y}_r, \vec{\pi}_r)$ in DA is unique, then $\vec{\pi}_r$ corresponds to an exact Nash equilibrium. Under $\epsilon > 0$ and $\vec{\pi}_{base}$, if the rest point $(\vec{y}_r, \vec{\pi}_r)$ in single-round DRDA is unique, and there exists a generalized Nash distribution in the POSG, then the NashConv of $\vec{\pi}_r$ is at most $\epsilon n \log \mathcal{K} \sum_t \gamma^t$, where $\mathcal{K} = \max\limits_{i \in N, x^i \in \mathcal{X}^i, a \in \mathcal{A}} \frac{1}{\pi_{base}^i(x^i, a)}$.*

Theorem 3 relates the DRDA rest point to Nash equilibrium under the assumption of its uniqueness. To find an approximate Nash equilibrium, we theoretically require the regularization parameter $\epsilon$ to be close to zero. If the POSG is (strictly) monotone, then by Theorem 2, we can use an infinitesimal $\epsilon$ in DRDA while still guaranteeing its convergence since we have $\lambda = 0$. In this case, a single round of DRDA is sufficient to find an arbitrarily precise approximate Nash equilibrium.

On the other hand, we can establish a more general guarantee for multi-round DRDA (Definition 3). Note that when the rest-point sequence $(p_l)_{l \geq 0}$ converges, the regularization terms approach zero at the rest points. Using Theorem 3, we further prove that the limit point of multi-round DRDA is an NE rather than a QRE under the uniqueness assumption.

**Theorem 4.** *If the policies in the rest-point sequence $(p_l)_{l \geq 0}$ generated by multi-round DRDA converge (under a properly selected $\epsilon > 0$ and $p_0$), and the rest point of DA is unique, then the corresponding policy sequence must converge to a Nash equilibrium.*

While Theorem 4 does not provide an exact condition for multi-round DRDA to converge, it suggests that an arbitrarily precise approximate NE can always be found as long as the overall learning process converges. In Appendix B.2, we also provide some evidence to show that the uniqueness assumption could be removed, e.g., when the game is individually concave. In Section 5, we will further show that multi-round DRDA actually converges to Nash equilibrium at a fast rate in a variety of games.

## 5 EXPERIMENTS

Considering the representation capability of the POSG framework, DRDA can be applied to a broad class of finite-action games. We implement DRDA in a range of games commonly formulated as NFGs, EFGs, MGs, and POSGs to verify the single-round/multi-round convergence and equilibrium-finding capability of DRDA. The implementation details, parameter settings, and game descriptions are placed in Appendices E.1, E.3, and F, respectively. Since the per-iteration time complexity of discrete-time DRDA (SDRDA; see Algorithm 1) is a standard $\mathcal{O}(|\mathcal{H}|)$ when we use dynamic programming to compute the advantage value, it is fair to compare its multi-round version (simply referred to as DRDA in the experiment) with other algorithms under the same total iterations.

Among the comparative algorithms, magnetic mirror descent (MMD), which exhibits last-iterate convergence to QREs in two-player zero-sum or (strictly) monotone games (Sokota et al., 2023),

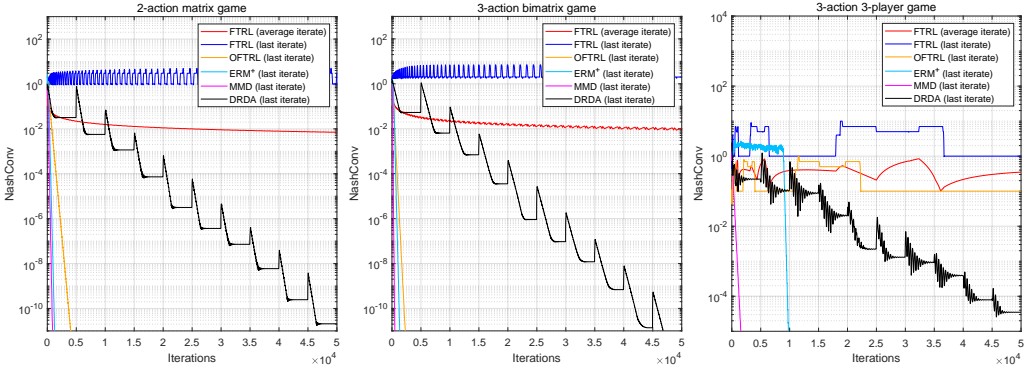

Figure 1: NashConv learning curves in NFG

has an intrinsic relationship with single-round DRDA if we consider another way of discretization (see Appendix E.2). Besides, an extended version of MMD uses the idea of moving magnets to find exact Nash equilibrium, while DRDA employs the idea of multi-round learning to achieve the same goal. Since MMD shares certain similarities with DRDA, we use its moving-magnet version (simply referred to as MMD in the experiment) as a common baseline for NFGs and EFGs.

## 5.1 NORMAL-FORM GAME (NFG)

NFGs are games where each of the players makes one decision simultaneously. As single-round DRDA can be viewed as a discounted variant of FTRL in NFGs, we compare it with vanilla FTRL and the FTRL variant under the optimistic gradient method (OFTRL for short; see Boone & Mertikopoulos (2024)). We also compare the variant of regret matching under the extragradient method (ERM+ for short; see Farina et al. (2024)) and use NashConv as the performance metric for all algorithms. The payoff settings for three NFG scenarios (i.e., 2-action matrix game, 3-action bimatrix game, and 3-action 3-player game) are provided in Appendix F.1, and the learning curves are shown in Figure 1.

In the 2-action matrix game and 3-action bimatrix game, while the average iterates of FTRL can converge to Nash equilibrium, its last iterates cannot. OFTRL as well as ERM+ and MMD, however, exhibit last-iterate convergence to NE at an exponential rate. For DRDA, the last iterates converge to a rest point in each single round separated by the vertical lines. This aligns with the linear convergence result in Theorem 2. Moreover, the rest-point policy sequence converges and approaches NE under multiple learning rounds. This is consistent with the statement in Theorem 4. Besides, the overall convergence rate is near-exponential since the curve drawn by the stationary policies is roughly linear.

In the 3-player NFG (with $3^3 = 27$ joint actions), MMD still converges, and ERM+ converges after an oscillation. However, the average iterate of FTRL and the last iterate of OFTRL no longer converge to NE. Actually, there is no theoretical guarantee for them to work in multiplayer games. In comparison, the single-round/multi-round convergence of DRDA is still guaranteed in this scenario.

## 5.2 EXTENSIVE-FORM GAME (EFG), MARKOV GAME (MG) & TYPICAL POSG

EFGs are games where each of the players makes multiple sequential decisions based on imperfect information about the global history. Since each player in an EFG may have multiple decision points, the size of the action space in the equivalent NFG representation can be exponential in the size of the game tree. Therefore, it is computationally impractical to directly apply some of the NFG-based methods to EFGs without considering the game-tree structure. Extending the idea of regret matching, counterfactual regret minimization (CFR) (Zinkevich et al., 2007) is the most commonly used equilibrium-learning algorithm in EFGs. Empirically, CFR+ (see (Bowling et al., 2015)) has an improved average-iterate/last-iterate performance. Regularized Nash dynamics (R-NaD for short; see Perolat et al. (2021; 2022)) is a multi-round learning algorithm that also exhibits last-iterate convergence in EFGs. Here, we compare CFR+, (stable) predictive CFR+ (PCFR+ for short; see Farina et al. (2021a; 2024)), extragradient CFR+ (ECFR+ for short), and R-NaD in multiplayer Kuhn poker scenarios. The game details are provided in Appendix F.2.

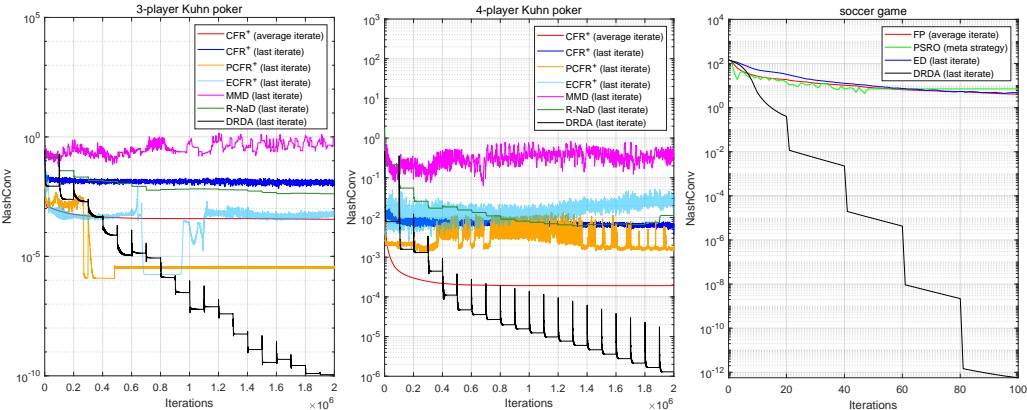

Figure 2: NashConv learning curves in (multiplayer) EFG and (infinite-horizon) MG

As is shown in Figure 2 (left & mid), while the average iterates of CFR$^+$ can converge, its last iterates oscillate. R-NaD has a multi-round learning pattern close to DRDA, but the process is much slower and suffers from an oscillation in the 4-player scenario. While ECFR$^+$ and MMD have impressive performance in NFGs, they do not work when it comes to multiplayer EFGs. PCFR$^+$ exhibits certain last-iterate convergence in the 3-player case but oscillates when it comes to 4 players. The 20-round DRDA clearly outperforms the other algorithms in 3-player/4-player Kuhn poker.

Markov games can be regarded as a special case of POSGs, where each player has full observation of the global state. The Markovian property guarantees that it is sufficient to use current states to represent histories and decision points. Since there are finite distinct decision points, tabular DRDA can solve infinite-horizon MGs like the soccer game (see Appendix F.3), where we compare DRDA with three other general methods of equilibrium learning: FP (Brown, 1951), PSRO (Lanctot et al., 2017), and ED (Lockhart et al., 2019). While both FP and PSRO require preserving history policies, ED only uses last-iterate policies like DRDA. Since the evaluation of value functions requires repeated dynamic programming in each iteration, we only run a total of 100 iterations (5 rounds for DRDA) to save time. As is shown in Figure 2 (right), while the 5-round DRDA has not converged in each single round of 20 iterations, the overall convergence is significantly faster than the comparative algorithms.

Besides EFG and MG, we also run multi-round DRDA in typical POSGs like the tiger games (see Wiggers (2015)) and find that it consistently outperforms the existing methods like heuristic search value iteration (HSVI) and sequence form linear program (SFLP). The comparison details are placed in Appendix F.4. Figure 5 provides the NashConv learning curves of 10-round DRDA under different regularization parameters $\epsilon \in \{0.05, 0.1, 0.15\}$ in Adversarial Tiger with time horizon $H \in \{2, 3, 4\}$.

## 6 CONCLUSION

In this paper, a multi-round equilibrium-learning system, based on discounted FTRL and named DRDA, is introduced under the framework of POSG. We define a generalized Nash distribution in POSGs and further show that the rest point of single-round DRDA can be characterized by this QRE concept (Theorem 1). We prove linear last-iterate convergence of DRDA to its rest point in a single round under the assumption of game hypomonotonicity (Theorem 2). We further prove that the limit point of multi-round DRDA must be a Nash equilibrium under the uniqueness assumption (Theorem 3 and Theorem 4). In our experiments, the discrete-time implementation of multi-round DRDA manages to approach Nash equilibrium in various games represented by POSGs. The last-iterate convergence of single-round/multi-round DRDA is consistent with theory. Besides, like magnetic mirror descent and extragradient regret matching, multi-round DRDA achieves a near-exponential rate in NFGs. For EFG, MG, and typical POSG, DRDA can outperform the existing baseline algorithms.

**Limitation & Future work.** One limitation of this work is that we have not theoretically examined the condition for the policy sequence of multi-round DRDA to converge despite the simulation results. It is also left for future work to examine whether the requirement of a unique equilibrium point can be removed in establishing the relationship to Nash equilibrium in multiplayer games.

ACKNOWLEDGMENTS

This work was supported in part by the National Natural Science Foundation of China under Grant 62293541 and Grant 62136008, in part by Beijing Natural Science Foundation under Grant 4232056, and in part by Beijing Nova Program under Grant 20240484514.

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

# A  LAST-ITERATE CONVERGENCE OF MORE FTRL VARIANTS

Besides discounted updates, the ideas of optimistic gradient methods and extragradient methods can also be employed to mitigate the recurrence behavior of FTRL and facilitate last-iterate convergence.

Note that the rest point of (1) is a $(\vec{y}_r, \vec{\pi}_r)$ pair, where $\vec{y}_r$ is a solution to the variational inequality problem (VIP) of $F(\vec{y}) = 0$ under $F(\vec{y}) = -\vec{w}(\vec{\sigma}(\vec{y}))$, and the discrete-time counterpart of (1) can be written as $\vec{y}_{t+1} = \vec{y}_t - \eta F(\vec{y}_t)$.

**Optimistic gradient method.**  Instead of using the current gradient $F(\vec{y}_t)$ to update, the optimistic gradient method uses $(2F(\vec{y}_t) - F(\vec{y}_{t-1}))$. The update formula then becomes $\vec{y}_{t+1} = \vec{y}_t - 2\eta F(\vec{y}_t) + \eta F(\vec{y}_{t-1})$. This method turns the original FTRL into the optimistic FTRL (OFTRL).

**Extragradient method.**  Instead of using $F(\vec{y}_t)$, the extragradient method uses the gradient at the point after one update attempt: $F(\vec{y}_t - \eta F(\vec{y}_t))$. The update formula then becomes $\vec{y}_{t+1} = \vec{y}_t - \eta F(\vec{y}_t - \eta F(\vec{y}_t))$. This turns the original FTRL into what we call extragradient FTRL (EFTRL).

In the field of optimization, both the optimistic gradient method and the extragradient method guarantee $\mathcal{O}(1/\mathcal{I})$ last-iterate convergence to the solution to the VIP under monotonicity assumptions (Gorbunov et al., 2022b;a), where $\mathcal{I}$ denotes iterations. For OFTRL, a near-optimal $\tilde{\mathcal{O}}(1)$ regret bound that corresponds to a $\tilde{\mathcal{O}}(1/\mathcal{I})$ convergence rate is also proved (Daskalakis et al., 2021), and last-iterate convergence to Nash equilibrium is guaranteed in two-player zero-sum games under the unique equilibrium assumption (Daskalakis & Panageas, 2019). For EFTRL, we find that it exhibits a learning behavior similar to OFTRL in various game scenarios.

**Reward regularization.**  Besides the gradient methods, Perolat et al. (2021; 2022) demonstrate that divergence regularization $-\frac{\eta}{\Pr(h_t|\pi^{-i})} \log \frac{\pi^i(x_t^i, a_t^i)}{\pi_{base}^i(x_t^i, a_t^i)}$ on rewards can lead to last-iterate convergence for FTRL. With this idea, a multi-round learning algorithm, regularized Nash dynamics (R-NaD), is also proposed to solve two-player zero-sum or monotone EFGs. As DRDA is also a multi-round learning system that is capable of dealing with EFGs, R-NaD is directly compared with multi-round DRDA in our experiment section.

## B  AUXILIARY RESULTS

### B.1  RELATED TO ACTION VALUES

The existing research on reinforcement learning has established a series of theoretical results when analyzing Markov decision processes (MDPs). As POSGs generalize MDPs by introducing multiple players and partial observations, some existing results related to action values in MDPs/POMDPs can also be generalized to POSGs. The following two lemmas will be used in our subsequent proofs.

First, the advantage function has a well-known property:

**Lemma 5** (Property of Advantage)**.**

$$\sum_{a_t^i} \pi^i(x_t^i, a_t^i) A_{\vec{\pi}}^i(h_t, a_t^i) = 0$$

*Proof.* By definition of the history-based advantage:

$$A_{\vec{\pi}}^i(h_t, a_t^i) = Q_{\vec{\pi}}^i(h_t, a_t^i) - V_{\vec{\pi}}^i(h_t)$$

Therefore, we have:

$$\sum_{a_t^i} \pi^i(x_t^i, a_t^i) A_{\vec{\pi}}^i(h_t, a_t^i)$$

$$= \sum_{a_t^i} \pi^i(x_t^i, a_t^i) \left( Q_{\vec{\pi}}^i(h_t, a_t^i) - V_{\vec{\pi}}^i(h_t) \right)$$

$$= \sum_{a_t^i} \pi^i(x_t^i, a_t^i) Q_{\vec{\pi}}^i(h_t, a_t^i) - V_{\vec{\pi}}^i(h_t) \sum_{a_t^i} \pi^i(x_t^i, a_t^i)$$

$$= V_{\vec{\pi}}^i(h_t) - V_{\vec{\pi}}^i(h_t) = 0$$

$\square$

Second, the well-known lemma of policy difference also holds in POSGs:

**Lemma 6** (Policy Difference)**.**

$$V_{\pi_\dagger^i, \vec{\pi}^{-i}}^i(h_t) - V_{\vec{\pi}}^i(h_t) = \mathbb{E}\left[ \sum_{k=t}^{T-1} \gamma^{k-t} A_{\vec{\pi}}^i(h_k, a_k^i) \,\Big|\, h_t, \pi_\dagger^i, \vec{\pi}^{-i} \right]$$

*Proof.* First, we have:

$$\sum_{a_k^i} \pi_\dagger^i(x_k^i, a_k^i) A_{\vec{\pi}}^i(h_k, a_k^i)$$

$$= \sum_{a_k^i} \pi_\dagger^i(x_k^i, a_k^i) Q_{\vec{\pi}}^i(h_k, a_k^i) - V_{\vec{\pi}}^i(h_k)$$

$$= \sum_{h_{k+1} \sqsupset h_k} \Pr\left(h_{k+1}|h_k, \pi_\dagger^i, \vec{\pi}^{-i}\right) \begin{pmatrix} R^i(s_k, \vec{a}_k, s_{k+1}) \\ + \gamma V_{\vec{\pi}}^i(h_{k+1}) \end{pmatrix} - V_{\vec{\pi}}^i(h_k)$$

Now we can expand the RHS expectation:

$$\mathbb{E}\left[\sum_{k=t}^{T-1}\gamma^{k-t}A_{\vec{\pi}}^{i}(h_{k},a_{k}^{i})\,\big|\,h_{t},\pi_{\dagger}^{i},\vec{\pi}^{-i}\right]$$

$$=\sum_{k=t}^{T-1}\gamma^{k-t}\mathbb{E}\left[A_{\vec{\pi}}^{i}(h_{k},a_{k}^{i})\,\big|\,h_{t},\pi_{\dagger}^{i},\vec{\pi}^{-i}\right]$$

$$=\sum_{k=t}^{T-1}\gamma^{k-t}\sum_{h_{k}\sqsupseteq h_{t}}\Pr\left(h_{k}|h_{t},\pi_{\dagger}^{i},\vec{\pi}^{-i}\right)\sum_{a_{k}^{i}}\pi_{\dagger}^{i}(x_{k}^{i},a_{k}^{i})A_{\vec{\pi}}^{i}(h_{k},a_{k}^{i})$$

$$=\sum_{k=t}^{T-1}\gamma^{k-t}\sum_{h_{k+1}\sqsupseteq h_{t}}\Pr\left(h_{k+1}|h_{t},\pi_{\dagger}^{i},\vec{\pi}^{-i}\right)R^{i}(s_{k},\vec{a}_{k},s_{k+1})$$

$$+\left(\begin{array}{c}\displaystyle\sum_{k=t+1}^{T-1}\gamma^{k-t}\sum_{h_{k}\sqsupseteq h_{t}}\Pr\left(h_{k}|h_{t},\pi_{\dagger}^{i},\vec{\pi}^{-i}\right)V_{\vec{\pi}}^{i}(h_{k})\\[2ex]\displaystyle-\sum_{k=t}^{T-1}\gamma^{k-t}\sum_{h_{k}\sqsupseteq h_{t}}\Pr\left(h_{k}|h_{t},\pi_{\dagger}^{i},\vec{\pi}^{-i}\right)V_{\vec{\pi}}^{i}(h_{k})\end{array}\right)$$

$$=V_{\pi_{\dagger}^{i},\vec{\pi}^{-i}}^{i}(h_{t})-V_{\vec{\pi}}^{i}(h_{t})$$

which derives the LHS value difference in the equality. $\square$

### B.2 RELATED TO INDIVIDUAL CONCAVITY

**Definition 8.** *A finite-action game is **individually concave** if for any $i \in N$, any $\vec{\pi}^{-i} \in \Pi^{-i}$, any $\pi_1^i, \pi_2^i \in \Pi^i$, and any $\alpha \in [0, 1]$:*

$$u^i((1-\alpha)\pi_1^i + \alpha\pi_2^i, \vec{\pi}^{-i}) \geq (1-\alpha)u^i(\pi_1^i, \vec{\pi}^{-i}) + \alpha u^i(\pi_2^i, \vec{\pi}^{-i})$$

Note that the individual concavity defined here follows the common definitions in games with a continuous action space (Bravo et al., 2018; Cai et al., 2022). In the finite-action setting that we consider, the policy space corresponds to the continuous action space. In the context of POSGs, $\pi^i = (1-\alpha)\pi_1^i + \alpha\pi_2^i$ means that $\pi^i(x^i, \cdot) = (1-\alpha)\pi_1^i(x^i, \cdot) + \alpha\pi_2^i(x^i, \cdot)$ at each decision point $x^i \in \mathcal{X}^i$. The individual concavity is well-defined in the sense that $\pi_1^i, \pi_2^i \in \Pi^i \Rightarrow \pi^i \in \Pi^i$. Also note that when $T = |S| = |O| = 1$, the POSG is reduced to an NFG, and the individual concavity always holds since $u^i(\pi^i, \vec{\pi}^{-i})$ is linear in the individual policy $\pi^i$ in NFGs.

Under the assumption of individual concavity, it is direct to prove the same NashConv bound as the one in Theorem 3 for the rest point of DA or single-round DRDA without relying on the uniqueness of the rest point (see Theorem 5). It is therefore mathematically correct as well to replace the *uniqueness assumption* in Theorem 4 with individual concavity. This further suggests the feasibility of applying single-round/multi-round DRDA to a broader class of games with multiple Nash equilibria.

**Theorem 5.** *If the POSG is individually concave, then every rest point $(\vec{y}_r, \vec{\pi}_r)$ in DA induces a Nash equilibrium $\vec{\pi}_r$, and every rest point $(\vec{y}_r, \vec{\pi}_r)$ in single-round DA (under $\epsilon > 0$ and $\vec{\pi}_{base}$) induces a joint policy $\vec{\pi}_r$ with NashConv at most $\epsilon n \log \mathcal{K} \sum_t \gamma^t$, where $\mathcal{K} = \max\limits_{i \in N, x^i \in \mathcal{X}^i, a \in \mathcal{A}} \frac{1}{\pi_{base}^i(x^i, a)}$.*

*Proof.* For any $i \in N, \pi^i \in \Pi^i$, and $\alpha \in (0, 1]$, define:

$$\vec{\pi}_{\dagger} = (\pi_{\dagger}^i, \vec{\pi}_r^{-i}) = ((1-\alpha)\pi_r^i + \alpha\pi^i, \vec{\pi}_r^{-i})$$

Using Lemma 6, we have:

$$V_{\pi_{\dagger}^{i},\vec{\pi}_{r}^{-i}}^{i}(h_{t})-V_{\vec{\pi}_{r}}^{i}(h_{t})=\mathbb{E}\left[\sum_{k=t}^{T-1}\gamma^{k-t}A_{\vec{\pi}_{r}}^{i}(h_{k},a_{k}^{i})\,\big|\,h_{t},\pi_{\dagger}^{i},\vec{\pi}_{r}^{-i}\right]$$

Therefore:

$$u^i(\vec{\pi}_\dagger) - u^i(\vec{\pi}_r) = \mathbb{E}_{h_0}\left[V^i_{\pi^i_\dagger, \vec{\pi}_r^{-i}}(h_0)\right] - \mathbb{E}_{h_0}\left[V^i_{\vec{\pi}_r}(h_0)\right]$$

$$= \mathbb{E}\left[\sum_{k=0}^{T-1}\gamma^k A^i_{\vec{\pi}_r}(h_k, a^i_k)\,\big|\,\pi^i_\dagger, \vec{\pi}_r^{-i}\right]$$

$$= \sum_{k=0}^{T-1}\gamma^k \sum_{h_k}\Pr\left(h_k|\pi^i_\dagger, \vec{\pi}_r^{-i}\right)\sum_{a^i_k}\pi^i_\dagger(x^i_k, a^i_k)A^i_{\vec{\pi}_r}(h_k, a^i_k)$$

By assumption on the individual concavity:

$$u^i(\vec{\pi}_\dagger) \geq (1-\alpha)u^i(\vec{\pi}_r) + \alpha u^i(\pi^i, \vec{\pi}_r^{-i})$$

Combining the two formulas above, we have:

$$\sum_{k=0}^{T-1}\gamma^k\sum_{h_k}\Pr\left(h_k|\pi^i_\dagger, \vec{\pi}_r^{-i}\right)\sum_{a^i_k}\pi^i_\dagger(x^i_k, a^i_k)A^i_{\vec{\pi}_r}(h_k, a^i_k) \geq \alpha\left(u^i(\pi^i, \vec{\pi}_r^{-i}) - u^i(\vec{\pi}_r)\right)$$

Using Lemma 5, we have:

$$\sum_{a^i_k}\pi^i_\dagger(x^i_k, a^i_k)A^i_{\vec{\pi}_r}(h_k, a^i_k) = \alpha\sum_{a^i_k}(\pi^i - \pi^i_r)(x^i_k, a^i_k)A^i_{\vec{\pi}_r}(h_k, a^i_k)$$

Therefore:

$$\sum_{k=0}^{T-1}\gamma^k\sum_{h_k}\Pr\left(h_k|\pi^i_\dagger, \vec{\pi}_r^{-i}\right)\sum_{a^i_k}(\pi^i - \pi^i_r)(x^i_k, a^i_k)A^i_{\vec{\pi}_r}(h_k, a^i_k) \geq u^i(\pi^i, \vec{\pi}_r^{-i}) - u^i(\vec{\pi}_r)$$

Letting $\alpha \to 0^+$, we have:

$$\sum_{k=0}^{T-1}\gamma^k\sum_{h_k}\Pr\left(h_k|\vec{\pi}_r\right)\sum_{a^i_k}(\pi^i - \pi^i_r)(x^i_k, a^i_k)A^i_{\vec{\pi}_r}(h_k, a^i_k) \geq u^i(\pi^i, \vec{\pi}_r^{-i}) - u^i(\vec{\pi}_r) \qquad (11)$$

Since $(\vec{y}_r, \vec{\pi}_r)$ is a rest point of DA ($\epsilon = 0$) or single-round DRDA ($\epsilon > 0$), we have:

$$\vec{y}_r = \vec{v}(\vec{\pi}_r) \Rightarrow \pi^i_r(x^i) \in \operatorname*{arg\,max}_{\pi^i(x^i)\in\Delta(\mathcal{A})}\left\{\langle\pi^i(x^i), v^i(\vec{\pi}_r)(x^i)\rangle - \epsilon D_{\mathrm{KL}}\left(\pi^i(x^i)||\pi^i_{base}(x^i)\right)\right\} \; (\forall i \in N, x^i \in \mathcal{X}^i)$$

By definition of $\vec{v}(\cdot)$ (3), it holds for any $i \in N, x^i \in \mathcal{X}^i$:

$$\sum_{h\in x^i}\Pr\left(h|\vec{\pi}_r\right)\left(\sum_{a^i}\pi^i_r(x^i, a^i)A^i_{\vec{\pi}_r}(h, a^i) - \epsilon D_{\mathrm{KL}}\left(\pi^i_r(x^i)||\pi^i_{base}(x^i)\right)\right)$$

$$\geq \sum_{h\in x^i}\Pr\left(h|\vec{\pi}_r\right)\left(\sum_{a^i}\pi^i(x^i, a^i)A^i_{\vec{\pi}_r}(h, a^i) - \epsilon D_{\mathrm{KL}}\left(\pi^i(x^i)||\pi^i_{base}(x^i)\right)\right)$$

With inequality (11), it is clear:

$$u^i(\pi^i, \vec{\pi}_r^{-i}) - u^i(\vec{\pi}_r) \leq \epsilon\sum_{k=0}^{T-1}\gamma^k\sum_{h_k}\Pr\left(h_k|\vec{\pi}_r\right)\left(D_{\mathrm{KL}}\left(\pi^i(x^i)||\pi^i_{base}(x^i)\right) - D_{\mathrm{KL}}\left(\pi^i(x^i)||\pi^i_{base}(x^i)\right)\right)$$

$$\leq \epsilon\sum_{k=0}^{T-1}\gamma^k\sum_{h_k}\Pr\left(h_k|\vec{\pi}_r\right)\log\mathcal{K} \leq \epsilon\log\mathcal{K}\sum_{k=0}^{T-1}\gamma^k$$

When $\epsilon = 0$, the joint policy $\vec{\pi}_r$ is clearly a Nash equilibrium. $\qquad\square$

## C OMITTED PROOFS

### C.1 PROOF OF LEMMA 1

*Proof.* Recall that the score $\vec{y}$ in single-round DRDA (4) has the following integral form:

$$y_t^i(x^i, a^i) = e^{-t} y_0^i(x^i, a^i) + \int_0^t e^{-(t-\tau)} v^i(\vec{\pi}_\tau)(x^i, a^i) d\tau$$

When the POSG has a finite horizon, the value $V_{\vec{\pi}}^i$ is bounded since the reward functions $\{R^i\}$ are bounded. When the POSG has an infinite horizon, the value $V_{\vec{\pi}}^i$ is still bounded since we have $0 < \gamma < 1$ and $\sum_{k=0}^{\infty} \gamma^k = \frac{1}{1-\gamma}$. Therefore, in either case, the advantage $A_{\vec{\pi}}^i$ and advantage value $v^i(\vec{\pi})$ are also bounded.

On the other hand, $\int_0^t e^{-(t-\tau)} d\tau = e^{\tau-t}\big|_0^t = 1 - e^{-t} \in [0, 1)$. Since $v^i(\vec{\pi}_\tau)$ is always bounded, with the basic property of integrals, we directly prove the boundedness of the score $\vec{y}$. □

### C.2 PROOF OF LEMMA 2

*Proof.* First, we prove:

$$\mu = \underset{\mu \in \Delta(\mathcal{A})}{\arg\max} \left\{ \sum_{a \in \mathcal{A}} \mu(a) \left( r(a) - \log \mu(a) \right) \right\} \Rightarrow \mu(a) \propto e^{r(a)}$$

Write the corresponding optimization problem:

$$\begin{cases} \text{maximize} \sum_{a \in \mathcal{A}} \mu(a) \left( r(a) - \log \mu(a) \right) \\ \text{s.t.} \quad \sum_{a \in \mathcal{A}} \mu(a) = 1 \\ \quad \mu(a) \geq 0, \ \forall a \in \mathcal{A} \end{cases}$$

Using the Lagrange multiplier, we have:

$$L = \sum_{a \in \mathcal{A}} \mu(a) \left( r(a) - \log \mu(a) \right) - \lambda \left( \sum_{a \in \mathcal{A}} \mu(a) - 1 \right)$$

$$\frac{\partial L}{\partial \mu(a)} = 0 \Rightarrow r(a) - \left( \log \mu(a) + \frac{\mu(a)}{\mu(a)} \right) - \lambda = 0$$

$$\Rightarrow \mu(a) = e^{r(a)-\lambda-1} \Rightarrow \mu(a) \propto e^{r(a)}$$

By definition of $\sigma^i(\cdot)$ (5), we have:

$$\pi_t^i(x^i) = \sigma^i(y_t^i)(x^i) = \underset{\pi^i(x^i) \in \Delta(\mathcal{A})}{\arg\max} \left\{ \begin{array}{l} \sum_{a^i \in \mathcal{A}} \pi^i(x^i, a^i) y_t^i(x^i, a^i) \\ - \epsilon D_{\text{KL}} \left( \pi^i(x^i) || \pi_{base}^i(x^i) \right) \end{array} \right\}$$

$$= \underset{\pi^i(x^i) \in \Delta(\mathcal{A})}{\arg\max} \left\{ \sum_{a^i \in \mathcal{A}} \pi^i(x^i, a^i) \left( \frac{1}{\epsilon} y_t^i(x^i, a^i) - \log \frac{\pi^i(x^i, a^i)}{\pi_{base}^i(x^i, a^i)} \right) \right\}$$

$$= \underset{\pi^i(x^i) \in \Delta(\mathcal{A})}{\arg\max} \left\{ \sum_{a^i \in \mathcal{A}} \pi^i(x^i, a^i) \left( \begin{array}{l} \frac{1}{\epsilon} y_t^i(x^i, a^i) + \log \pi_{base}^i(x^i, a^i) \\ - \log \pi^i(x^i, a^i) \end{array} \right) \right\}$$

Therefore:

$$\pi_t^i(x^i, a^i) \propto \exp\left( \frac{1}{\epsilon} y_t^i(x^i, a^i) + \log \pi_{base}^i(x^i, a^i) \right) = \pi_{base}^i(x^i, a^i) \exp\left( \frac{1}{\epsilon} y_t^i(x^i, a^i) \right)$$

□

## C.3 PROOF OF LEMMA 3

*Proof.* For any $i \in N$ and any own information set $x_t^i$, define $\vec{\pi}_\dagger = (\pi_\dagger^i, \pi_*^{-i})$. Let $\vec{\pi}_\ddagger$ be a joint policy that equals $\vec{\pi}_*$ initially and switches to $\vec{\pi}_\dagger$ after player $i$ reaches $x_t^i$. By definition of GND:

$$\mathbb{E}_{h_0}\left[V_{\vec{\pi}_*}^i(h_0)\right] - \epsilon \sum_{k=0}^{T-1} \gamma^k \sum_{x_k^i} \left(D_{\mathrm{KL}}\left(\pi_*^i(x_k^i)||\pi_{base}^i(x_k^i)\right) \sum_{h_k \in x_k^i} \Pr\left(h_k|\vec{\pi}_*\right)\right)$$

$$\geq \mathbb{E}_{h_0}\left[V_{\vec{\pi}_\ddagger}^i(h_0)\right] - \epsilon \sum_{k=0}^{T-1} \gamma^k \sum_{x_k^i} \left(D_{\mathrm{KL}}\left(\pi_\ddagger^i(x_k^i)||\pi_{base}^i(x_k^i)\right) \sum_{h_k \in x_k^i} \Pr\left(h_k|\vec{\pi}_*\right)\right)$$

For both sides, eliminate the terms related to the histories outside of the subtrees whose roots belong to $x_t^i$, we have:

$$\sum_{h_t \in x_t^i} \Pr\left(h_t|\vec{\pi}_*\right) V_{\vec{\pi}_*}^i(h_t) - \epsilon \sum_{k=t}^{T-1} \gamma^{k-t} \sum_{x_k^i \sqsupseteq x_t^i} \left(D_{\mathrm{KL}}\left(\pi_*^i(x_k^i)||\pi_{base}^i(x_k^i)\right) \sum_{h_k \in x_k^i} \Pr\left(h_k|\vec{\pi}_*\right)\right)$$

$$\geq \sum_{h_t \in x_t^i} \Pr\left(h_t|\vec{\pi}_*\right) V_{\vec{\pi}_\dagger}^i(h_t) - \epsilon \sum_{k=t}^{T-1} \gamma^{k-t} \sum_{x_k^i \sqsupseteq x_t^i} \left(D_{\mathrm{KL}}\left(\pi_\dagger^i(x_k^i)||\pi_{base}^i(x_k^i)\right) \sum_{h_k \in x_k^i} \Pr\left(h_k|\vec{\pi}_*\right)\right)$$

For any $\pi^i \in \Pi^i$ and $\alpha \in [0,1]$, we let $\pi_\dagger^i = (1-\alpha)\pi_*^i + \alpha\pi^i$ and define:

$$g(\alpha) = \left(\sum_{h_t \in x_t^i} \Pr\left(h_t|\vec{\pi}_*\right) V_{\vec{\pi}_\dagger}^i(h_t) - \epsilon \sum_{k=t}^{T-1} \gamma^{k-t} \sum_{x_k^i \sqsupseteq x_t^i} \left(D_{\mathrm{KL}}\left(\pi_\dagger^i(x_k^i)||\pi_{base}^i(x_k^i)\right) \sum_{h_k \in x_k^i} \Pr\left(h_k|\vec{\pi}_*\right)\right)\right)$$

$$- \left(\sum_{h_t \in x_t^i} \Pr\left(h_t|\vec{\pi}_*\right) V_{\vec{\pi}_*}^i(h_t) - \epsilon \sum_{k=t}^{T-1} \gamma^{k-t} \sum_{x_k^i \sqsupseteq x_t^i} \left(D_{\mathrm{KL}}\left(\pi_*^i(x_k^i)||\pi_{base}^i(x_k^i)\right) \sum_{h_k \in x_k^i} \Pr\left(h_k|\vec{\pi}_*\right)\right)\right)$$

Clearly, $g(0) = 0$ and $\forall \alpha \in (0,1], g(\alpha) \leq 0$, which yields:

$$\nabla g(\alpha)|_{\alpha \to 0^+} \leq 0 \tag{12}$$

Using Lemma 6, we have:

$$V_{\vec{\pi}_\dagger}^i(h_t) - V_{\vec{\pi}_*}^i(h_t) = \sum_{k=t}^{T-1} \gamma^{k-t} \sum_{h_k \sqsupseteq h_t} \Pr\left(h_t \to h_k|\vec{\pi}_\dagger\right) \sum_{a_k^i} \pi_\dagger^i(x_k^i, a_k^i) A_{\vec{\pi}_*}^i(h_k, a_k^i)$$

Therefore:

$$g(\alpha) = \sum_{h_t \in x_t^i} \Pr\left(h_t|\vec{\pi}_*\right) \sum_{k=t}^{T-1} \gamma^{k-t} \sum_{h_k \sqsupseteq h_t} \left(\Pr\left(h_t \to h_k|\vec{\pi}_\dagger\right) \sum_{a_k^i} \pi_\dagger^i(x_k^i, a_k^i) A_{\vec{\pi}_*}^i(h_k, a_k^i)\right) -$$

$$\epsilon \sum_{k=t}^{T-1} \gamma^{k-t} \sum_{x_k^i \sqsupseteq x_t^i} \left(\left(D_{\mathrm{KL}}\left(\pi_\dagger^i(x_k^i)||\pi_{base}^i(x_k^i)\right) - D_{\mathrm{KL}}\left(\pi_*^i(x_k^i)||\pi_{base}^i(x_k^i)\right)\right) \sum_{h_k \in x_k^i} \Pr\left(h_k|\vec{\pi}_*\right)\right)$$

Using Lemma 5, we have:

$$\sum_{a_k^i} \pi_\dagger^i(x_k^i, a_k^i) A_{\vec{\pi}_*}^i(h_k, a_k^i) = \alpha \sum_{a_k^i} (\pi^i - \pi_*^i)(x_k^i, a_k^i) A_{\vec{\pi}_*}^i(h_k, a_k^i)$$

Then, we compute the derivative:

$$\nabla g(\alpha) = \sum_{h_t \in x_t^i} \Pr(h_t|\vec{\pi}_*) \sum_{k=t}^{T-1} \gamma^{k-t} \sum_{h_k \sqsupseteq h_t} \left( \begin{array}{l} \nabla \Pr(h_t \to h_k|\vec{\pi}_\dagger) \sum_{a_k^i} \pi_\dagger^i(x_k^i, a_k^i) A_{\vec{\pi}_*}^i(h_k, a_k^i) + \\ \Pr(h_t \to h_k|\vec{\pi}_\dagger) \sum_{a_k^i} (\pi^i - \pi_*^i)(x_k^i, a_k^i) A_{\vec{\pi}_*}^i(h_k, a_k^i) \end{array} \right)$$

$$- \epsilon \sum_{k=t}^{T-1} \gamma^{k-t} \sum_{x_k^i \sqsupseteq x_t^i} \left( \sum_{a_k^i} (\pi^i - \pi_*^i)(x_k^i, a_k^i) \left( \log \pi_\dagger^i(x_k^i, a_k^i) - \log \pi_{base}^i(x_k^i, a_k^i) \right) \sum_{h_k \in x_k^i} \Pr(h_k|\vec{\pi}_*) \right)$$

Let $\alpha \to 0^+$ and simplify the equality:

$$\nabla g(\alpha)|_{\alpha \to 0^+} = \sum_{h_t \in x_t^i} \sum_{k=t}^{T-1} \gamma^{k-t} \sum_{h_k \sqsupseteq h_t} \Pr(h_t|\vec{\pi}_*) \sum_{a_k^i} (\pi^i - \pi_*^i)(x_k^i, a_k^i) A_{\vec{\pi}_*}^i(h_k, a_k^i)$$

$$- \epsilon \sum_{k=t}^{T-1} \gamma^{k-t} \sum_{x_k^i \sqsupseteq x_t^i} \left( \sum_{a_k^i} (\pi^i - \pi_*^i)(x_k^i, a_k^i) \left( \begin{array}{l} \log \pi_*^i(x_k^i, a_k^i) - \\ \log \pi_{base}^i(x_k^i, a_k^i) \end{array} \right) \sum_{h_k \in x_k^i} \Pr(h_k|\vec{\pi}_*) \right)$$

Note that the following inequality always holds:

$$\sum_{a_k^i} \pi^i(x_k^i, a_k^i) \log \pi_*^i(x_k^i, a_k^i) \le \sum_{a_k^i} \pi^i(x_k^i, a_k^i) \log \pi^i(x_k^i, a_k^i)$$

Therefore, with the inequality (12), we have:

$$\sum_{k=t}^{T-1} \gamma^{k-t} \sum_{\substack{x_k \sqsupseteq x_t \\ h_k \in x_k}} \Pr(h_k|\vec{\pi}_*) \sum_{a_k^i} (\pi^i - \pi_*^i)(x_k^i, a_k^i) A_{\vec{\pi}_*}^i(h_k, a_k^i)$$

$$\le \sum_{k=t}^{T-1} \gamma^{k-t} \sum_{\substack{x_k \sqsupseteq x_t \\ h_k \in x_k}} \Pr(h_k|\vec{\pi}_*) \cdot \epsilon \left( \begin{array}{l} D_{\mathrm{KL}}\left( \pi^i(x_k^i)||\pi_{base}^i(x_k^i) \right) - \\ D_{\mathrm{KL}}\left( \pi_*^i(x_k^i)||\pi_{base}^i(x_k^i) \right) \end{array} \right)$$

Let $\pi^i(x_k^i) = \pi_*^i(x_k^i)$ for all $x_k^i$ under $k \ge t+1$ and simplify the inequality:

$$\sum_{h_t \in x_t^i} \Pr(h_t|\vec{\pi}_*) \left( \sum_{a_t^i} \pi_*^i(x_t^i, a_t^i) A_{\vec{\pi}_*}^i(h_t, a_t^i) - \epsilon D_{\mathrm{KL}}\left( \pi_*^i(x_t^i)||\pi_{base}^i(x_t^i) \right) \right)$$

$$\ge \sum_{h_t \in x_t^i} \Pr(h_t|\vec{\pi}_*) \left( \sum_{a_t^i} \pi^i(x_t^i, a_t^i) A_{\vec{\pi}_*}^i(h_t, a_t^i) - \epsilon D_{\mathrm{KL}}\left( \pi^i(x_t^i)||\pi_{base}^i(x_t^i) \right) \right)$$

$\square$

## C.4 PROOF OF THEOREM 1

*Proof.* Let $\vec{y}_t = \vec{v}(\vec{\pi}_*)$ and $\vec{\pi}_t = \vec{\pi}_*$. Compute (5) using the definition of $\vec{v}(\cdot)$ (3):

$$\arg\max_{\pi^i(x^i)\in\Delta(\mathcal{A})} \left\{ \langle \pi^i(x^i), y_t^i(x^i) \rangle - \epsilon D_{\mathrm{KL}}\left(\pi^i(x^i)||\pi_{base}^i(x^i)\right) \right\}$$

$$= \arg\max_{\pi^i(x^i)\in\Delta(\mathcal{A})} \left\{ \sum_{a^i\in\mathcal{A}} \pi^i(x^i,a^i) \frac{\sum_{h\in x^i} \Pr(h|\vec{\pi}_*) A_{\vec{\pi}_*}^i(h,a^i)}{\sum_{h\in x^i} \Pr(h|\vec{\pi}_*)} - \epsilon D_{\mathrm{KL}}\left(\pi^i(x^i)||\pi_{base}^i(x^i)\right) \right\}$$

$$= \arg\max_{\pi^i(x^i)\in\Delta(\mathcal{A})} \left\{ \sum_{h\in x^i} \Pr(h|\vec{\pi}_*) \left( \sum_{a^i\in\mathcal{A}} \pi^i(x^i,a^i) A_{\vec{\pi}_*}^i(h,a^i) - \epsilon D_{\mathrm{KL}}\left(\pi^i(x^i)||\pi_{base}^i(x^i)\right) \right) \right\}$$

By Lemma 3, for any $i\in N, \pi^i\in\Pi^i, x^i\in\mathcal{X}^i$:

$$\sum_{h\in x^i} \Pr(h|\vec{\pi}_*) \left( \sum_{a^i\in\mathcal{A}} \pi_*^i(x^i,a^i) A_{\vec{\pi}_*}^i(h_t,a^i) - \epsilon D_{\mathrm{KL}}\left(\pi_*^i(x^i)||\pi_{base}^i(x^i)\right) \right)$$

$$\geq \sum_{h\in x^i} \Pr(h|\vec{\pi}_*) \left( \sum_{a^i\in\mathcal{A}} \pi^i(x^i,a^i) A_{\vec{\pi}_*}^i(h_t,a^i) - \epsilon D_{\mathrm{KL}}\left(\pi^i(x^i)||\pi_{base}^i(x^i)\right) \right)$$

For $\epsilon > 0$, it is clear that $\vec{\pi}_t = \vec{\pi}_* = \vec{\sigma}(\vec{y}_t)$.

For $\epsilon = 0$, it is clear that $\pi_t^i(x^i) \in \arg\max_{\pi^i(x^i)\in\Delta(\mathcal{A})} \left\{ \langle \pi^i(x^i), y_t^i(x^i) \rangle \right\}, \forall x^i \in \mathcal{X}^i$.

Since $\dot{\vec{y}}_t = \vec{v}(\vec{\pi}_t) - \vec{y}_t = \vec{v}(\vec{\pi}_*) - \vec{v}(\vec{\pi}_*) = 0$, we prove that $(\vec{v}(\vec{\pi}_*), \vec{\pi}_*)$ is a rest point of single-round DRDA when $\epsilon > 0$ and is a rest point of DA when $\epsilon = 0$. $\qquad\square$

## C.5 PROOF OF LEMMA 4

*Proof.* With respect to $\ell_1$ norm $\|\cdot\|_1$, we show that $D_{\mathrm{KL}}(\pi||\mu) = \sum_a \pi(a)\log\frac{\pi(a)}{\mu(a)}$ is a 1-strongly convex function of $\pi$ under a fixed $\mu$, using the widely known result that the negative Gibbs entropy $H(\pi) = \sum_a \pi(a)\log\pi(a)$ is 1-strongly convex (***Example 2*** in Shalev-Shwartz & Singer (2006)).

By definition of strong convexity:

$$H(\alpha\pi + (1-\alpha)\pi') \leq \alpha H(\pi) + (1-\alpha)H(\pi') - \frac{1}{2}\alpha(1-\alpha)\|\pi - \pi'\|_1^2$$

Since $H(\pi) = D_{\mathrm{KL}}(\pi||\mu) + \sum_a \pi(a)\log\mu(a)$, we have:

$$D_{\mathrm{KL}}(\alpha\pi + (1-\alpha)\pi'||\mu) + \sum_a (\alpha\pi + (1-\alpha)\pi')(a)\log\mu(a)$$

$$\leq \alpha D_{\mathrm{KL}}(\pi||\mu) + \alpha\sum_a \pi(a)\log\mu(a)$$

$$+ (1-\alpha)D_{\mathrm{KL}}(\pi'||\mu) + (1-\alpha)\sum_a \pi'(a)\log\mu(a)$$

$$- \frac{1}{2}\alpha(1-\alpha)\|\pi - \pi'\|_1^2$$

Simplifying the inequality, we have:

$$D_{\mathrm{KL}}(\alpha\pi + (1-\alpha)\pi'||\mu) \leq \alpha D_{\mathrm{KL}}(\pi||\mu) + (1-\alpha)D_{\mathrm{KL}}(\pi'||\mu) - \frac{1}{2}\alpha(1-\alpha)\|\pi - \pi'\|_1^2$$

which means that $D_{\mathrm{KL}}(\pi||\mu)$ is also 1-strongly convex with respect to $\ell_1$ norm $\|\cdot\|_1$. $\qquad\square$

## C.6 PROOF OF THEOREM 2

*Proof.* Construct a Fenchel coupling:

$$F^i(\pi^i, y^i) = \sum_{k=0}^{T-1} \sum_{h_k} \max_{\pi_{\dagger}^i(x_k^i) \in \Delta(\mathcal{A})} \left\{ \left\langle \pi_{\dagger}^i(x_k^i), y^i(x_k^i) \right\rangle - \epsilon D_{\mathrm{KL}} \left( \pi_{\dagger}^i(x_k^i) || \pi_{base}^i(x_k^i) \right) \right\}$$

$$- \sum_{k=0}^{T-1} \sum_{h_k} \left( \left\langle \pi^i(x_k^i), y^i(x_k^i) \right\rangle - \epsilon D_{\mathrm{KL}} \left( \pi^i(x_k^i) || \pi_{base}^i(x_k^i) \right) \right)$$

By Lemma 4, $\epsilon D_{\mathrm{KL}} \left( \pi^i(x_k^i) || \pi_{base}^i(x_k^i) \right)$ is an $\epsilon$-strongly convex function of $\pi^i(x_k^i)$ with respect to $\ell_1$ norm. Besides, it is continuous on $\Delta(\mathcal{A})$. Therefore, we have the following inequality (***Proposition 4.3*** in Mertikopoulos & Zhou (2019)):

$$\max_{\pi_{\dagger}^i(x_k^i) \in \Delta(\mathcal{A})} \left\{ \left\langle \pi_{\dagger}^i(x_k^i), y^i(x_k^i) \right\rangle - \epsilon D_{\mathrm{KL}} \left( \pi_{\dagger}^i(x_k^i) || \pi_{base}^i(x_k^i) \right) \right\}$$

$$- \left( \left\langle \pi^i(x_k^i), y^i(x_k^i) \right\rangle - \epsilon D_{\mathrm{KL}} \left( \pi^i(x_k^i) || \pi_{base}^i(x_k^i) \right) \right)$$

$$\geq \frac{1}{2} \epsilon \left\| \sigma^i(y^i)(x_k^i) - \pi^i(x_k^i) \right\|_1^2$$

$$\geq \frac{1}{2} \epsilon \left\| \sigma^i(y^i)(x_k^i) - \pi^i(x_k^i) \right\|_2^2$$

Enumerating all $(k, h_k)$ pairs and summing up the corresponding terms for both sides, we have:

$$F^i(\pi^i, y^i) \geq \frac{1}{2} \epsilon \left\| \sigma^i(y^i) - \pi^i \right\|_2^2 \tag{13}$$

Recall that $\sigma^i(\cdot)$ has an equivalent expression (7). Therefore, the corresponding *max* function is continuously differentiable with respect to $y^i$. Applying the envelope theorem (***Theorem 1.F.1*** in Takayama & Akira (1985)), we have:

$$\nabla_{y^i(x_k^i)} \left( \max_{\pi_{\dagger}^i(x_k^i) \in \Delta(\mathcal{A})} \left\{ \left\langle \pi_{\dagger}^i(x_k^i), y^i(x_k^i) \right\rangle - \epsilon D_{\mathrm{KL}} \left( \pi_{\dagger}^i(x_k^i) || \pi_{base}^i(x_k^i) \right) \right\} \right) = \sigma^i(y^i)(x_k^i)$$

On the other hand, we have the derivative result:

$$\nabla_{y^i(x_k^i)} \left( \left\langle \pi^i(x_k^i), y^i(x_k^i) \right\rangle - \epsilon D_{\mathrm{KL}} \left( \pi^i(x_k^i) || \pi_{base}^i(x_k^i) \right) \right) = \pi^i(x_k^i)$$

Therefore, we have:

$$\nabla_{y^i} F^i(\pi^i, y^i) = \sigma^i(y^i) - \pi^i$$

For single-round DRDA (4), define a continuously differentiable energy function:

$$\mathcal{V}(\vec{y}_t) = \sum_{i \in N} F^i(\pi_{\epsilon}^i, y_t^i)$$

Compute its time derivative:

$$\dot{\mathcal{V}}(\vec{y}_t) = \sum_{i \in N} \left\langle \nabla_{y^i} F^i(\pi_{\epsilon}^i, y_t^i), \dot{y}_t^i \right\rangle$$

$$= \sum_{i \in N} \left\langle \sigma^i(y_t^i) - \pi_{\epsilon}^i, v^i(\vec{\pi}_t) - y_t^i \right\rangle$$

$$= \sum_{i \in N} \left( \left\langle \pi_t^i - \pi_{\epsilon}^i, v^i(\vec{\pi}_t) \right\rangle - \left\langle \pi_t^i - \pi_{\epsilon}^i, y_t^i \right\rangle \right)$$

where $\vec{\pi}_t = \vec{\sigma}(\vec{y}_t)$.

Since $\vec{\pi}_\epsilon = \vec{\sigma}(\vec{y}_\epsilon)$, by definition of $\vec{\sigma}(\cdot)$ (5) and the Fenchel coupling, we have:

$$\left\langle \pi_t^i - \pi_\epsilon^i, y_t^i \right\rangle = F^i(\pi_\epsilon^i, y_t^i) + \epsilon \sum_{k=0}^{T-1} \sum_{h_k} \left( D_{\mathrm{KL}}\left( \pi_t^i(x_k^i) \| \pi_{base}^i(x_k^i) \right) - D_{\mathrm{KL}}\left( \pi_\epsilon^i(x_k^i) \| \pi_{base}^i(x_k^i) \right) \right)$$

$$\left\langle \pi_\epsilon^i - \pi_t^i, y_\epsilon^i \right\rangle = F^i(\pi_t^i, y_\epsilon^i) + \epsilon \sum_{k=0}^{T-1} \sum_{h_k} \left( D_{\mathrm{KL}}\left( \pi_\epsilon^i(x_k^i) \| \pi_{base}^i(x_k^i) \right) - D_{\mathrm{KL}}\left( \pi_t^i(x_k^i) \| \pi_{base}^i(x_k^i) \right) \right)$$

Summing up the two equations above, we have:

$$\left\langle \pi_t^i - \pi_\epsilon^i, y_t^i \right\rangle + \left\langle \pi_\epsilon^i - \pi_t^i, y_\epsilon^i \right\rangle = F^i(\pi_\epsilon^i, y_t^i) + F^i(\pi_t^i, y_\epsilon^i)$$

which implies:

$$\dot{\mathcal{V}}(\vec{y}_t) = \sum_{i \in N} \left( \left\langle \pi_t^i - \pi_\epsilon^i, v^i(\vec{\pi}_t) - y_\epsilon^i \right\rangle - F^i(\pi_\epsilon^i, y_t^i) - F^i(\pi_t^i, y_\epsilon^i) \right)$$

Since $(\vec{y}_\epsilon, \vec{\pi}_\epsilon)$ is a rest point of single-round DRDA, we have:

$$\dot{\mathcal{V}}(\vec{y}_t) = \sum_{i \in N} \left\langle \pi_t^i - \pi_\epsilon^i, v^i(\vec{\pi}_t) - v^i(\vec{\pi}_\epsilon) \right\rangle - \sum_{i \in N} F^i(\pi_\epsilon^i, y_t^i) - \sum_{i \in N} F^i(\pi_t^i, y_\epsilon^i)$$

Using the assumption on the local hypomonotonicity (10) and the inequality (13), it holds in $\Omega$:

$$\dot{\mathcal{V}}(\vec{y}_t) \leq \lambda \sum_{i \in N} \left\| \pi_t^i - \pi_\epsilon^i \right\|_2^2 - 2 \cdot \frac{1}{2}\epsilon \sum_{i \in N} \left\| \pi_t^i - \pi_\epsilon^i \right\|_2^2 = (\lambda - \epsilon) \sum_{i \in N} \left\| \pi_t^i - \pi_\epsilon^i \right\|_2^2 \leq 0$$

Using inequality (13), we further have:

$$\dot{\mathcal{V}}(\vec{y}_t) \leq -\frac{2(\epsilon - \lambda)}{\epsilon} \mathcal{V}(\vec{y}_t) = -2(1 - \frac{\lambda}{\epsilon})\mathcal{V}(\vec{y}_t)$$

Note that $\mathcal{V}(\vec{y}_t)$ is non-negative. Therefore, with $\epsilon > \lambda$, we have:

$$\mathcal{V}(\vec{y}_t) \leq \mathcal{V}(\vec{y}_0) \exp\left( -2(1 - \frac{\lambda}{\epsilon})t \right)$$

which means $\mathcal{V}(\vec{y}_t)$ approaches zero exponentially fast.

Also note that by (13), $\mathcal{V}(\vec{y}) = 0 \Rightarrow \sum_{i \in N} \left\| \sigma^i(y^i) - \pi_\epsilon^i \right\|_2^2 = 0 \Rightarrow \vec{\pi} = \vec{\sigma}(\vec{y}) = \vec{\pi}_\epsilon$.

As $\Omega$ is a compact set positively invariant with respect to (4), we prove that for a single-round DRDA starting from an arbitrary $\vec{y}_0 \in \Omega$, the policy always converges to $\vec{\pi}_\epsilon$ at a linear rate. $\qquad\square$

## C.7 PROOF OF PROPOSITION 1

*Proof.* Consider two joint policies $(\pi^1, \cdots, \pi^j, \cdots, \pi^n)$ and $(\pi^1, \cdots, \pi^j_\dagger, \cdots, \pi^n)$ that differ only in player $j$'s policy. We hypothesize that there always exists a constant $L^i_j$ such that:

$$\left\| v^i(\pi^1, \cdots, \pi^j, \cdots, \pi^n) - v^i(\pi^1, \cdots, \pi^j_\dagger, \cdots, \pi^n) \right\|_2 \le L^i_j \left\| \pi^j - \pi^j_\dagger \right\|_2 \tag{14}$$

Now we verify this hypothesis under $T = |S| = |O| = 1$. In this case, the POSG is equivalent to an NFG, with $v^i(\vec{\pi})(x^i_0, a) = w^i(\vec{\pi})(a) - u^i(\vec{\pi})$ for the unique decision point $x^i_0$, where $w^i(\vec{\pi})(a) = u^i(\pi^i_a, \vec{\pi}^{-i})$ is player $i$'s expected utility under a pure strategy $\pi^i_a(\cdot)$ with $\pi^i_a(a) = 1$.

Let $\vec{\pi} = (\pi^1, \cdots, \pi^j, \cdots, \pi^n), \vec{\pi}_\dagger = (\pi^1, \cdots, \pi^j_\dagger, \cdots, \pi^n), L = \max_{\vec{\pi} \in \Pi} \left| u^i(\vec{\pi}) \right|$, and we have:

$$\left\| v^i(\vec{\pi}) - v^i(\vec{\pi}_\dagger) \right\|_2 \le \left\| v^i(\vec{\pi}) - v^i(\vec{\pi}_\dagger) \right\|_1 = \left\| v^i(\vec{\pi})(x^i_0) - v^i(\vec{\pi}_\dagger)(x^i_0) \right\|_1$$
$$\le \left\| w^i(\vec{\pi}) - w^i(\vec{\pi}_\dagger) \right\|_1 + |\mathcal{A}| \cdot \left| u^i(\vec{\pi}) - u^i(\vec{\pi}_\dagger) \right|$$

When $i = j$:

$$\left\| w^i(\vec{\pi}) - w^i(\vec{\pi}_\dagger) \right\|_1 = 0$$
$$\left| u^i(\vec{\pi}) - u^i(\vec{\pi}_\dagger) \right| = \left| \sum_{a \in \mathcal{A}} u^j(\pi^j_a, \vec{\pi}^{-j}) \left( \pi^j(a) - \pi^j_\dagger(a) \right) \right| \le L \left\| \pi^j - \pi^j_\dagger \right\|_1$$

When $i \ne j$:

$$\left\| w^i(\vec{\pi}) - w^i(\vec{\pi}_\dagger) \right\|_1 = \sum_{a \in \mathcal{A}} \left| u^i(\pi^i_a, \vec{\pi}^{-i}) - u^i(\pi^i_a, \vec{\pi}^{-i}_\dagger) \right| \le |\mathcal{A}| \max_{a \in \mathcal{A}} \left| u^i(\pi^i_a, \vec{\pi}^{-i}) - u^i(\pi^i_a, \vec{\pi}^{-i}_\dagger) \right|$$

$$\left| u^i(\vec{\pi}) - u^i(\vec{\pi}_\dagger) \right| = \left| \sum_{a \in \mathcal{A}} \pi^i(a) \left( u^i(\pi^i_a, \vec{\pi}^{-i}) - u^i(\pi^i_a, \vec{\pi}^{-i}_\dagger) \right) \right| \le \max_{a \in \mathcal{A}} \left| u^i(\pi^i_a, \vec{\pi}^{-i}) - u^i(\pi^i_a, \vec{\pi}^{-i}_\dagger) \right|$$

Let $\vec{\pi}_1 = (\pi^i_a, \vec{\pi}^{-i}), \vec{\pi}_2 = (\pi^i_a, \vec{\pi}^{-i}_\dagger)$, and we have:

$$\left| u^i(\pi^i_a, \vec{\pi}^{-i}) - u^i(\pi^i_a, \vec{\pi}^{-i}_\dagger) \right| = \left| u^i(\vec{\pi}_1) - u^i(\vec{\pi}_2) \right| = \left| \sum_{b \in \mathcal{A}} u^i(\pi^j_b, \vec{\pi}^{-j}_1) \left( \pi^j_1(b) - \pi^j_2(b) \right) \right| \le L \left\| \pi^j - \pi^j_\dagger \right\|_1$$

Therefore, it always holds:

$$\left\| w^i(\vec{\pi}) - w^i(\vec{\pi}_\dagger) \right\|_1 + |\mathcal{A}| \cdot \left| u^i(\vec{\pi}) - u^i(\vec{\pi}_\dagger) \right| \le 2 |\mathcal{A}| L \left\| \pi^j - \pi^j_\dagger \right\|_1 \le 2|\mathcal{A}|^{\frac{3}{2}} L \left\| \pi^j - \pi^j_\dagger \right\|_2$$

where the last step follows Cauchy-Schwarz inequality.

Setting $L^i_j = 2|\mathcal{A}|^{\frac{3}{2}} L = 2|\mathcal{A}|^{\frac{3}{2}} \max_{\vec{\pi} \in \Pi} \left| u^i(\vec{\pi}) \right|$, we can see that the inequality (14) holds. With this inequality, we can decompose the inner product $\left\langle \pi^i_1 - \pi^i_2, v^i(\vec{\pi}_1) - v^i(\vec{\pi}_2) \right\rangle$ for two arbitrary joint policies $\vec{\pi}_1 = (\pi^i_1)_{i \in N}$ and $\vec{\pi}_2 = (\pi^i_2)_{i \in N}$. Specifically, we have:

$$\left\langle \pi^i_1 - \pi^i_2, v^i(\vec{\pi}_1) - v^i(\vec{\pi}_2) \right\rangle$$
$$= \left\langle \pi^i_1 - \pi^i_2, v^i(\pi^1_1, \cdots, \pi^n_1) - v^i(\pi^1_2, \pi^2_1, \cdots, \pi^n_1) \right\rangle$$
$$+ \left\langle \pi^i_1 - \pi^i_2, v^i(\pi^1_2, \pi^2_1, \cdots, \pi^n_1) - v^i(\pi^1_2, \pi^2_2, \pi^3_1, \cdots, \pi^n_1) \right\rangle$$
$$+ \cdots \cdots$$
$$+ \left\langle \pi^i_1 - \pi^i_2, v^i(\pi^1_2, \cdots, \pi^{n-1}_2, \pi^n_1) - v^i(\pi^1_2, \cdots, \pi^n_2) \right\rangle$$
$$\le \sum_{j \in N} \left\| \pi^i_1 - \pi^i_2 \right\|_2 L^i_j \left\| \pi^j_1 - \pi^j_2 \right\|_2$$

where the last step follows Cauchy-Schwarz inequality and (14).

Setting $\lambda = \max\limits_{i \in N} \left\{ \sum\limits_{j \in N} \frac{1}{2}(L_j^i + L_i^j) \right\}$, we have:

$$\sum_{i \in N} \left\langle \pi_1^i - \pi_2^i, v^i(\vec{\pi}_1) - v^i(\vec{\pi}_2) \right\rangle$$

$$\leq \sum_{i \in N} \sum_{j \in N} L_j^i \left\| \pi_1^i - \pi_2^i \right\|_2 \left\| \pi_1^j - \pi_2^j \right\|_2$$

$$\leq \sum_{i \in N} \sum_{j \in N} \frac{1}{2} L_j^i \left( \left\| \pi_1^i - \pi_2^i \right\|_2^2 + \left\| \pi_1^j - \pi_2^j \right\|_2^2 \right)$$

$$= \sum_{i \in N} \left( \sum_{j \in N} \frac{1}{2}(L_j^i + L_i^j) \right) \left\| \pi_1^i - \pi_2^i \right\|_2^2$$

$$\leq \lambda \sum_{i \in N} \left\| \pi_1^i - \pi_2^i \right\|_2^2$$

Thus, we prove that every NFG is globally $\lambda$-hypomonotone, where:

$$\lambda = \max_{i \in N} \left\{ \sum_{j \in N} |\mathcal{A}|^{\frac{3}{2}} \left( \max_{\vec{\pi} \in \Pi} \left| u^i(\vec{\pi}) \right| + \max_{\vec{\pi} \in \Pi} \left| u^j(\vec{\pi}) \right| \right) \right\}$$

$$= |\mathcal{A}|^{\frac{3}{2}} \sum_{j \in N} \left( \max_{i \in N, \vec{\pi} \in \Pi} \left| u^i(\vec{\pi}) \right| + \max_{\vec{\pi} \in \Pi} \left| u^j(\vec{\pi}) \right| \right)$$

$\square$

## C.8 PROOF OF THEOREM 3

*Proof.* By Theorem 1, every generalized Nash distribution under $\epsilon > 0$ or $\epsilon = 0$ induces a rest point in single-round DRDA or DA, respectively. If there exists a GND $\vec{\pi}_r$ that induces the unique rest point $(\vec{y}_r, \vec{\pi}_r)$ in single-round DRDA, then $\vec{\pi}_r$ must be the unique GND under the corresponding $\epsilon > 0$ and $\vec{\pi}_{base}$. Since Nash equilibrium always exists, if $(\vec{y}_r, \vec{\pi}_r)$ is the unique DA rest point, then $\vec{\pi}_r$ must be the unique GND under $\epsilon = 0$ (i.e., the unique NE).

Therefore, by Definition 4, for any $i \in N$ and $\pi^i \in \Pi^i$:

$$u^i(\pi^i, \vec{\pi}_r^{-i}) - u^i(\vec{\pi}_r) \leq$$

$$\epsilon \sum_{t=0}^{T-1} \gamma^t \sum_{x_t^i} \left( \left( D_{\mathrm{KL}}\left( \pi^i(x_t^i) || \pi_{base}^i(x_t^i) \right) - D_{\mathrm{KL}}\left( \pi_*^i(x_t^i) || \pi_{base}^i(x_t^i) \right) \right) \sum_{h_t \in x_t^i} \Pr\left( h_t | \vec{\pi}_* \right) \right)$$

For the non-negative KL-divergence terms, we let $\mathcal{K} = \max\limits_{i \in N, x^i \in \mathcal{X}^i, a \in \mathcal{A}} \frac{1}{\pi_{base}^i(x^i, a)}$ and further have:

$$D_{\mathrm{KL}}\left( \pi^i(x_t^i) || \pi_{base}^i(x_t^i) \right) = \sum_{a \in \mathcal{A}} \pi^i(x_t^i, a) \log \pi^i(x_t^i, a) - \sum_{a \in \mathcal{A}} \pi^i(x_t^i, a) \log \pi_{base}^i(x_t^i, a) \leq$$

$$-\sum_{a \in \mathcal{A}} \pi^i(x_t^i, a) \log \pi_{base}^i(x_t^i, a) = \sum_{a \in \mathcal{A}} \pi^i(x_t^i, a) \log \frac{1}{\pi_{base}^i(x_t^i, a)} \leq \sum_{a \in \mathcal{A}} \pi^i(x_t^i, a) \log \mathcal{K} = \log \mathcal{K}$$

Therefore:

$$u^i(\pi^i, \vec{\pi}_r^{-i}) - u^i(\vec{\pi}_r) \leq \epsilon \sum_{t=0}^{T-1} \gamma^t \sum_{x_t^i} \left( \log \mathcal{K} \sum_{h_t \in x_t^i} \Pr\left( h_t | \vec{\pi}_* \right) \right)$$

$$= \epsilon \log \mathcal{K} \sum_{t=0}^{T-1} \gamma^t \sum_{x_t^i} \sum_{h_t \in x_t^i} \Pr\left( h_t | \vec{\pi}_* \right)$$

Note that $\sum_{x_t^i} \sum_{h_t \in x_t^i} \Pr\left( h_t | \vec{\pi}_* \right) = \sum_{h_t} \Pr\left( h_t | \vec{\pi}_* \right) = 1, \forall t \geq 0$. Therefore, we have:

$$\mathrm{NashConv}(\vec{\pi}_r) = \sum_{i=1}^{n} \max_{\pi^i \in \Pi^i} \left\{ u^i(\pi^i, \vec{\pi}_r^{-i}) - u^i(\vec{\pi}_r) \right\} \leq \epsilon n \log \mathcal{K} \sum_{t=0}^{T-1} \gamma^t$$

Clearly, when $(\vec{y}_r, \vec{\pi}_r)$ is the unique rest point of DA, it corresponds to the case of $\epsilon = 0$, which implies that $\vec{\pi}_r$ is an exact Nash equilibrium. □

## C.9 PROOF OF THEOREM 4

*Proof.* By Definition 3, the rest points $p_l = (\vec{y}_l, \vec{\pi}_l)$ under different $l$ in multi-round DRDA correspond to the solutions to $\vec{y} = \vec{v}(\vec{\pi})$ under different policy selections.

Let $\vec{\pi}_\epsilon$ be the limit of the policy sequence $(\vec{\pi}_l)_{l \geq 0}$ when $l \to \infty$. Then, we have:

$$\lim_{l \to \infty} D_{\mathrm{KL}}\left( \pi_l^i(x^i) || \pi_{l-1}^i(x^i) \right) = D_{\mathrm{KL}}\left( \pi_\epsilon^i(x^i) || \pi_\epsilon^i(x^i) \right) = 0$$

Therefore, as the policy sequence converges, the regularization term $-\epsilon D_{\mathrm{KL}}\left( \pi^i(x^i) || \pi_{base}^i(x^i) \right)$ in (5) approaches zero at the rest point $(\vec{v}(\vec{\pi}), \vec{\pi})$. As a result, $(\vec{v}(\vec{\pi}_\epsilon), \vec{\pi}_\epsilon)$ corresponds to a rest point of DA. By Theorem 3, the limit policy $\vec{\pi}_\epsilon$ is a Nash equilibrium when DA has a unique rest point. □

# D  GAME HYPOMONOTONICITY

Since hypomonotonicity (see Gao & Pavel (2021); Gadjov & Pavel (2023)) is less commonly used in the existing literature of game theory than in the field of optimization, here we provide numerical simulations to show that it can be practically used to characterize multiplayer dynamic games.

The original Kuhn poker has two players with three ranks. Here we numerically test the global hypomonotonicity of its multiplayer variants, 3-player Kuhn poker with 5 ranks and 4-player Kuhn poker with 6 ranks. Both games are also used as test environments in our experiment section.

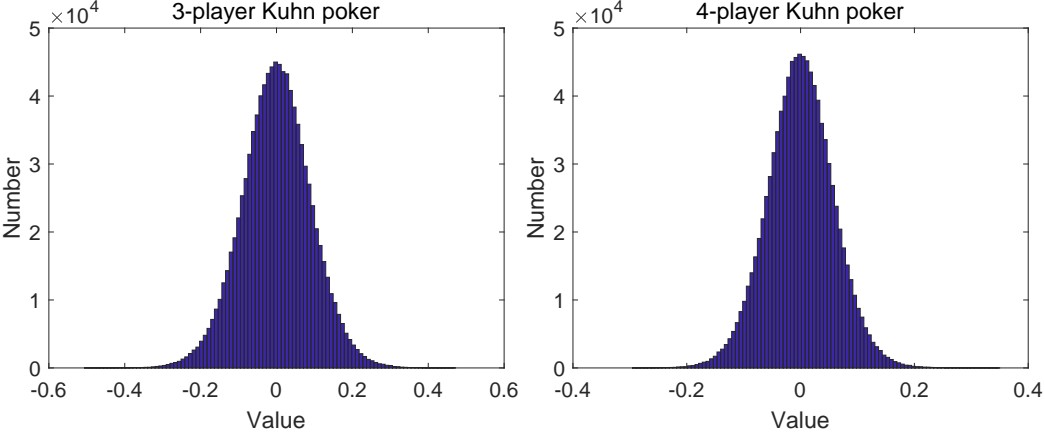

Figure 3: Histograms of sampled $\dfrac{\sum\limits_{i\in N}\left\langle\pi_1^i-\pi_2^i,v^i(\vec{\pi}_1)-v^i(\vec{\pi}_2)\right\rangle}{\sum\limits_{i\in N}\left\|\pi_1^i-\pi_2^i\right\|_2^2}$ in multiplayer Kuhn poker

By Definition 7, the supremum of the computed value $\dfrac{\sum\limits_{i\in N}\left\langle\pi_1^i-\pi_2^i,v^i(\vec{\pi}_1)-v^i(\vec{\pi}_2)\right\rangle}{\sum\limits_{i\in N}\left\|\pi_1^i-\pi_2^i\right\|_2^2}$ can be an estimate of a global hypomonotonicity value $\lambda$ and thus reflect its order of magnitude when the sampling is sufficient. Therefore, we compute the value by randomly sampling $10^6$ joint policy pairs $(\vec{\pi}_1,\vec{\pi}_2)$. The histograms of the sampled value are shown in Figure 3.

For 3-player Kuhn poker (with 5 ranks), the computed value is in the range of $[-0.505869, 0.472063]$.

For 4-player Kuhn poker (with 6 ranks), the computed value is in the range of $[-0.295345, 0.349964]$.

Note that the maximum value is at the level of $0.1$, which implies that the order of magnitude of a global $\lambda$ should not be too large. Therefore, it is safe to say the local hypomonotonicity assumption used in our convergence analysis is realistic in multiplayer Kuhn poker. For more game scenarios, we show the results of the maximum and minimum values under the same number of samples in Table 2.

Table 2: Maximum and minimum sampled values in more game scenarios

| Game Scenario | Maximum Value | Minimum Value |
|---|---|---|
| 2-player Kuhn poker with 3 ranks | 1.872479 | $-1.260188$ |
| 2-player Kuhn poker with 4 ranks | 0.861661 | $-0.824744$ |
| 3-player Kuhn poker with 4 ranks | 0.679113 | $-0.575718$ |
| 2-player Leduc poker with 2 suits and 3 ranks | 0.022138 | $-0.019163$ |
| 3-player Leduc poker with 2 suits and 3 ranks | 0.002436 | $-0.003294$ |
| 3-player Leduc poker with 3 suits and 3 ranks | 0.068833 | $-0.060182$ |

The maximum values provide estimations for the order of magnitude of the global $\lambda$. By Theorem 2, as long as $\lambda$ is a finite quantity, we can always guarantee the convergence of single-round DRDA by setting a sufficiently large $\epsilon$. On the other hand, Theorems 3 and 5 suggest that a larger $\epsilon$ leads to a larger NE gap for the rest-point policy. This can be viewed as a trade-off in the selection of $\epsilon$.

## E  IMPLEMENTATION DETAILS

### E.1  DISCRETE-TIME ALGORITHMS

We use the discrete-time counterpart of (4) as a practical implementation of single-round DRDA. An iteration variable $m$ is in place of the continuous time variable $t$. With the introduction of a step size $\eta$, the single-round algorithm can be viewed as an Euler method that solves the original ODE, and an iteration is exactly one step. The pseudocode for single-round DRDA is shown in Algorithm 1.

---

**Algorithm 1:** Single-round DRDA (SDRDA)

---

**Input:** Initial score $\vec{y}_0$, initial policy $\vec{\pi}_0$, iteration number $M$, and regularization parameter $\epsilon$

1  Set $\vec{\pi}_{base} = \vec{\pi}_0$
2  **for** $m \in \{0, 1, \cdots, M-1\}$ **do**
3  $\quad$ Compute all $\Pr(h|\vec{\pi}_m)$ and $A^i_{\vec{\pi}_m}(h, a^i)$ for all $i \in N$ (using dynamic programming)
4  $\quad$ **for** $i \in N$ **do**
5  $\quad\quad$ **for** $x^i \in \mathcal{X}^i$ **do**
6  $\quad\quad\quad$ **for** $a^i \in \mathcal{A}$ **do**
7  $\quad\quad\quad\quad$ Compute $v^i(\vec{\pi}_m)(x^i, a^i) = \dfrac{\sum_{h \in x^i} \Pr(h|\vec{\pi}_m) A^i_{\vec{\pi}_m}(h, a^i)}{\sum_{h \in x^i} \Pr(h|\vec{\pi}_m)}$
8  $\quad\quad\quad\quad$ Update $y^i_{m+1}(x^i, a^i) = y^i_m(x^i, a^i) + \eta\left(v^i(\vec{\pi}_m)(x^i, a^i) - y^i_m(x^i, a^i)\right)$
9  $\quad\quad\quad$ **end**
10 $\quad\quad\quad$ **for** $a^i \in \mathcal{A}$ **do**
11 $\quad\quad\quad\quad$ Update $\pi^i_{m+1}(x^i, a^i) = \dfrac{\pi^i_{base}(x^i, a^i) \exp\left(\frac{1}{\epsilon} y^i_{m+1}(x^i, a^i)\right)}{\sum_{b \in \mathcal{A}} \pi^i_{base}(x^i, b) \exp\left(\frac{1}{\epsilon} y^i_{m+1}(x^i, b)\right)}$
12 $\quad\quad\quad$ **end**
13 $\quad\quad$ **end**
14 $\quad$ **end**
15 **end**

**Output:** Last-iterate $(\vec{y}_M, \vec{\pi}_M)$

---

For the multi-round algorithm, we use SDRDA (Algorithm 1) as the oracle $\mathcal{M}$ in Definition 3. The pseudocode for multi-round DRDA is shown in Algorithm 2.

---

**Algorithm 2:** Multi-round DRDA (MDRDA)

---

**Input:** Round number $L$, iteration number $M$, and regularization parameter $\epsilon$

1  Initialize $\vec{y}_0$ and $\vec{\pi}_0$
2  **for** $l \in \{0, 1, \cdots, L-1\}$ **do**
3  $\quad$ $(\vec{y}_{l+1}, \vec{\pi}_{l+1}) = \text{SDRDA}(\vec{y}_l, \vec{\pi}_l, M, \epsilon)$
4  **end**

**Output:** Last-iterate policy $\vec{\pi}_L$

---

### E.2  RELATIONSHIP WITH MAGNETIC MIRROR DESCENT

Magnetic mirror descent (MMD) (Sokota et al., 2023) is a discrete-time learning dynamic based on mirror descent. Like single-round DRDA, MMD exhibits last-iterate convergence to QRE. Since mirror descent itself can be converted to a special form of FTRL by introducing a score $y$, here we try to mathematically compare MMD with DRDA.

For MMD, if we set the mirror map to be negative entropy and the step size to be 1, it has the closed-form update formula $\pi_{t+1} \propto (\pi_t \rho^\alpha e^{q_t})^{\frac{1}{1+\alpha}}$, where $\pi_t$ is the policy at discrete time $t$, $\rho$ is the magnet policy, $q_t$ is the Q-value, and $\alpha$ is the regularization temperature. Note that $\rho$ and $q_t$ in MMD are analogous to the base policy $\pi_{base}$ and the advantage value $v$ in DRDA, respectively. When $\rho$ is set to be a uniform policy, we have $\pi_{t+1} \propto (\pi_t e^{q_t})^{\frac{1}{1+\alpha}}$. Through recursion, we can write $\pi_{t+1} \propto e^{y_{t+1}}$, where $y_{t+1} = \sum_{k=1}^t (\frac{1}{1+\alpha})^{t+1-k} q_t$.

For DRDA, if we also set $\pi_{base}$ to be uniform, it can be generally written as $\pi_{t+1} \propto e^{y_{t+1}}$ in discrete time. If we further consider a form of discretization based on the integral formula (6) rather than the original ODE (4) and use $q_t$ to replace $v(\pi_t)$, we directly have $y_{t+1} = \sum_{k=1}^{t}(\frac{1}{e})^{t+1-k}q_t$. Clearly, this score is the same as that of MMD when we enforce $1 + \alpha = e$. Thus, DRDA can be related to MMD through a specific discretization if we ignore the base policy or the magnet.

Now we use a common notation $\rho$ to indicate the base policy in DRDA. When $\rho$ moves along time $t$ and is not kept as a fixed uniform policy, however, the update formula $\pi_{t+1} \propto \rho_t e^{y_{t+1}}$ for DRDA is quite different from the update formula $\pi_{t+1} \propto \rho_t^{\frac{\alpha}{1+\alpha}}(\pi_t e^{q_t})^{\frac{1}{1+\alpha}}$ for moving-magnet MMD. The above-mentioned relationship can no longer be established. That is to say, when employing a moving magnet, the behavior of DRDA will be completely different from MMD.

From a theoretical perspective, DRDA guarantees last-iterate convergence to QRE in all games with certain hypomonotonicity, while MMD is restricted to two-player zero-sum or (strictly) monotone games. When considering finding an exact Nash equilibrium, multi-round DRDA theoretically requires the base policy $\rho = \pi_{base}$ to be a fixed rest-point policy in each round. In comparison, moving-magnet MMD (see Appendix H.5 of Sokota et al. (2023)) directly learns in a single round under a changing $\rho = \rho_t$, and the related paper does not provide theoretical guarantees for this choice.

### E.3 PARAMETER SETTINGS

In Table 3, we show the parameter settings in our experiments on the discrete-time DRDA from Appendix E.1. Under the parameter settings, each single run of DRDA can be finished within one hour using a single Intel Core i7-12700F CPU.

Table 3: Parameter settings of DRDA

|  | matrix game | bimatrix game | 3-player game | Kuhn-3 | Kuhn-4 | soccer game |
|---|---|---|---|---|---|---|
| $L$ | 10 | 10 | 10 | 20 | 20 | 5 |
| $M$ | 5000 | 5000 | 5000 | $1 \times 10^5$ | $1 \times 10^5$ | 20 |
| $\epsilon$ | 0.1 | 0.1 | 0.3 | 0.02 | 0.015 | 0.1 |
| $\eta$ | 0.005 | 0.005 | 0.005 | 0.001 | 0.001 | 0.002 |

In principle, an arbitrary regularization parameter $\epsilon$ can be used as long as it is greater than a hypomonotonicity threshold $\lambda$ for the game (see Theorem 2). However, our experimental results show that a smaller gap for the rest point of single-round DRDA generally guarantees better performance of multi-round DRDA. Therefore, $\epsilon$ should not be too large as well (see Theorem 3). The setting above provides a moderate choice for $\epsilon$ and guarantees the performance of DRDA when the step size $\eta$ is sufficiently small (no greater than the level of 0.01).

In Table 4, we compare the NashConv of DRDA under different $\eta$. When $\eta$ is too large, DRDA will no longer converge (marked as "$-$"). Also note that when $\eta$ is overly small, the result can be not as good since DRDA has not reached its rest point in each single round under the same number of iterations. In each game scenario, a range of $\eta$ can guarantee good performance of DRDA.

Table 4: Last-iterate NashConv of DRDA under different $\eta$

| $\eta$ | matrix | bimatrix | 3-player | Kuhn-3 | Kuhn-4 | soccer |
|---|---|---|---|---|---|---|
| 0.001 | $1.4 \times 10^{-9}$ | $1.9 \times 10^{-9}$ | 0.014 | $1.1 \times 10^{-10}$ | $1.3 \times 10^{-6}$ | $6.9 \times 10^{-8}$ |
| 0.002 | $2.0 \times 10^{-11}$ | $1.6 \times 10^{-12}$ | $2.0 \times 10^{-4}$ | $1.1 \times 10^{-10}$ | $1.3 \times 10^{-6}$ | $5.7 \times 10^{-13}$ |
| 0.005 | $2.1 \times 10^{-11}$ | $1.2 \times 10^{-12}$ | $3.5 \times 10^{-5}$ | $6.0 \times 10^{-9}$ | $-$ | $9.8 \times 10^{-15}$ |
| 0.01 | $2.1 \times 10^{-11}$ | $-$ | $3.5 \times 10^{-5}$ | $-$ | $-$ | $1.6 \times 10^{-15}$ |
| 0.1 | $-$ | $-$ | $-$ | $-$ | $-$ | $-$ |

## F  EXPERIMENT DETAILS

### F.1  NORMAL-FORM GAME (NFG)

The payoff matrices of the 2-action matrix game (zero-sum) and the 3-action bimatrix game are shown in Tables 5 and 6, respectively.

Table 5: Payoff matrix of 2-action matrix game

|  | Player 2, Action 1 | Player 2, Action 2 |
|---|---|---|
| Player 1, Action 1 | $(1, -1)$ | $(0, 0)$ |
| Player 1, Action 2 | $(-2, 2)$ | $(3, -3)$ |

Table 6: Payoff matrix of 3-action bimatrix game

|  | Player 2, Action 1 | Player 2, Action 2 | Player 2, Action 3 |
|---|---|---|---|
| Player 1, Action 1 | $(9, 6)$ | $(3, 7)$ | $(6, 0)$ |
| Player 1, Action 2 | $(7, 0)$ | $(8, 2)$ | $(4, 4)$ |
| Player 1, Action 3 | $(2, 4)$ | $(1, 6)$ | $(3, 7)$ |

The payoff tensor of the 3-action 3-player game is shown in Table 7.

Table 7: Payoff matrix of 3-action 3-player game

|  | Player 3, Action 1 | Player 3, Action 2 | Player 3, Action 3 |
|---|---|---|---|
| Player 1&2, Action (1,1) | $(9, 5, 5)$ | $(8, 3, 8)$ | $(3, 5, 9)$ |
| Player 1&2, Action (1,2) | $(9, 6, 4)$ | $(3, 5, 7)$ | $(4, 10, 6)$ |
| Player 1&2, Action (1,3) | $(10, 6, 4)$ | $(6, 3, 0)$ | $(8, 0, 1)$ |
| Player 1&2, Action (2,1) | $(1, 4, 1)$ | $(0, 1, 1)$ | $(1, 2, 8)$ |
| Player 1&2, Action (2,2) | $(8, 3, 6)$ | $(5, 5, 6)$ | $(6, 0, 9)$ |
| Player 1&2, Action (2,3) | $(4, 2, 0)$ | $(1, 5, 8)$ | $(9, 5, 0)$ |
| Player 1&2, Action (3,1) | $(4, 9, 9)$ | $(9, 9, 7)$ | $(5, 1, 8)$ |
| Player 1&2, Action (3,2) | $(6, 5, 9)$ | $(10, 10, 0)$ | $(8, 3, 9)$ |
| Player 1&2, Action (3,3) | $(6, 3, 9)$ | $(1, 2, 2)$ | $(9, 4, 5)$ |

### F.2  EXTENSIVE-FORM GAME (EFG)

Since Kuhn poker (Kuhn, 1950) is a benchmark game from the extensive-form game literature, we use its multiplayer variants as EFG test environments. For 3-player Kuhn poker, we increase the number of ranks from 3 to 5. For 4-player Kuhn poker, we further increase the number of ranks to 6.

### F.3  MARKOV GAME (MG)

We use a two-player zero-sum soccer game as the test environment for infinite-horizon MG. Figure 4 is an illustration of the game. The two players are marked with A and B. The player who keeps the ball is marked with a cycle. Each player can choose an action from "up", "down", "left", "right", and "stay" at each time step. If the two players collide after the simultaneous move, then the ball possession exchanges. When the ball carrier moves into the opponent's goal, the game terminates. The winning player receives a reward of $+100$ and the opponent receives a reward of $-100$. The initial state distribution $\rho$ is set to be a uniform distribution, and the discount factor $\gamma$ is set to be $0.95$.

### F.4  TIGER GAME (TYPICAL POSG)

Adversarial Tiger and Competitive Tiger are two typical POSGs introduced by Wiggers (2015). We further test multi-round DRDA in the two games. For comparison purposes, we run a 5-round DRDA

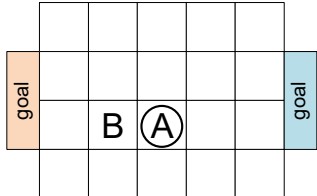

Figure 4: Illustration of soccer game

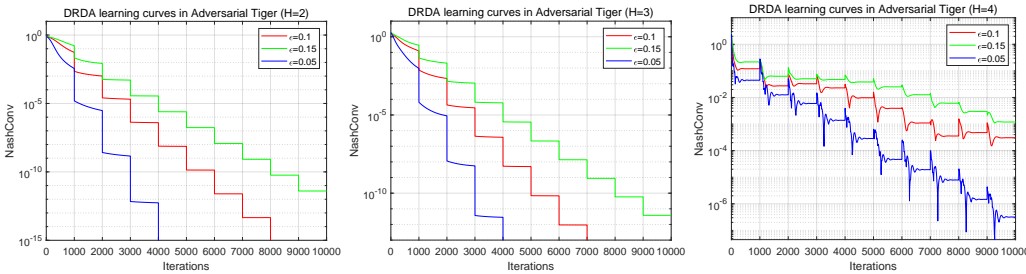

Figure 5: NashConv learning curves in Adversarial Tiger

($10^3$ iterations per round) under different time horizons $H \in \{2,3,4\}$ (using a fixed parameter setting $\epsilon = 0.1, \eta = 0.01$ for each game) and record the last-iterate NashConv as well as the time consumed.

According to the existing results (reported in Table 2 of Delage et al. (2024)), among heuristic search value iteration (HSVI), random search, informed search, sequence form linear program (SFLP), and CFR$^+$ algorithms, SFLP (Koller et al., 1994) demonstrates the best performance in the tiger games under $H \in \{2,3,4\}$. However, DRDA clearly outperforms SFLP when we use NashConv as an aligned metric to describe the reported results.

Table 8: Performance comparison between SFLP and DRDA in tiger games

| Game Scenario | SFLP (the best reported) | | DRDA (tested) | |
|---|---|---|---|---|
| | NashConv | Time | NashConv | Time |
| Adv Tiger $H = 2$ | 0.16 | < 1 sec | $< \mathbf{10^{-4}}$ | < 1 sec |
| Adv Tiger $H = 3$ | 0.24 | < 1 sec | $< \mathbf{10^{-4}}$ | < 1 sec |
| Adv Tiger $H = 4$ | 0.32 | 8 sec | $\mathbf{0.017}(0.010)$ | 8(10) sec |
| Comp Tiger $H = 2$ | 0.24 | < 1 sec | $< \mathbf{10^{-4}}$ | < 1 sec |
| Comp Tiger $H = 3$ | 0.36 | 48 sec | $\mathbf{0.028}$ | $\mathbf{3}$ sec |
| Comp Tiger $H = 4$ | 0.48 | 14 min | $\mathbf{0.024}$ | $\mathbf{5}$ min |

Note that in Adversarial Tiger with $H = 4$, the 5-round DRDA actually terminates using 10 seconds, but we also show the NashConv achieved in 8 seconds (without parentheses) for comparison purposes.

