# OpenReview forum: "Divergence-Regularized Discounted Aggregation: Equilibrium Finding in Multiplayer Partially Observable Stochastic Games"
_ICLR.cc/2025/Conference — ICLR 2025 Poster_

### Official Review · Reviewer_LUwx · 2024-10-24

**Soundness:** 2
**Presentation:** 2
**Contribution:** 2
**Rating:** 8
**Confidence:** 2

**Summary:**

The submission studies a regularized variant of FTRL for solving for equilibria. The submission derives continuous-time limit point results for convergence to Nash equilibria in general-sum settings and compares experimentally to a variety of baselines.

**Strengths:**

Converging to Nash in general-sum settings is a very difficult problem. Let me also say that I did not attempt to go through the submission's math. I evaluated the submission's soundness, presentation, and contribution as "fair" largely because I lack expertise to judge the novelty or correctness of the technical contribution. Similarly, I evaluated the submission as a 6 as I feel I lack the expertise to verify its value. Should other reviewers possess greater expertise I will defer to them, though I do feel confident saying that the writing is unclear in important places and the experimental results are weak. I'm also a bit concerned about the tension between the complexity results for computing equilibria in general-sum games and the claims made in the submission.

**Weaknesses:**

> solving partially observable stochastic games (POSGs), which unify normal-form games (NFGs), extensive-form games (EFGs),
and Markov games (MGs).

This isn't correct. In contrast to EFGs, POSGs can neither express untimeable games nor imperfect-recall games.

> While a wide range of game-theoretic learning dynamics, including no-regret dynamics and best response dynamics, were primarily analyzed in static normal-form games (NFG), many real-world games are dynamic and thus require to be solved under a different game representation

Inconsistent tense. Also, the reason they require a different representation is because treating them as NFGs is too expensive, not because they involve state.

>  imperfect information game

Requires hyphen.

>  For example, the perfect information game of Go can be formulated as a Markov game (MG) by ignoring the action
of the waiting player. The imperfect information game of Texas Hold’em is commonly formulated as
an extensive-form game (EFG). In view of this requirement,

These sentences don't flow correctly.

> As an extension of MGs, partially observable stochastic games (POSGs) introduces imperfect information and is capable of unifying NFGs, EFGs, and MGs.

Again, not true re EFGs.

> forizon

> The existence of the rest point of single-round DRDA is thus proved.

It's not clear to me why this is a non-trivial result.

> Theorem 1. In a POSG, every GND ⃗π∗ under ϵ > 0 induces a rest point (⃗v(⃗π∗), ⃗π∗) in single-round
DRDA, and every GND ⃗π∗ under ϵ = 0 (i.e., Nash equilibrium) induces a DA rest point (⃗v(⃗π∗), ⃗π∗).

This also seems kind of trivial to me.

---

Other weaknesses:
- The results are continuous time, not discrete time.
- The games studied in the experiments seem quite limited.

**Questions:**

> On the other hand,

On the other hand compared to what?

> For general-sum games, the last iterate of a “discounted” variant of FTRL, first examined in Leslie & Collins (2005), is proved to converge to the solution concept of Nash distribution (Coucheney et al., 2015; Gao & Pavel, 2021), a specific form of quantal
response equilibrium (QRE) (McKelvey & Palfrey, 1995)) defined in NFGs.

What's the catch here? Does this require exponential time for small quantal shocks?

>  Therefore, the hypomonotonicity assumption used in our convergence analysis should not be regarded as a strong
assumption.

Is this claim based on experimental results for Kuhn poker? That would strike me as far from comprehensive enough to make such a sweeping claim.

> These underestimates could reflect the order of magnitude of the true global hypomonotonicity value
λ, which must be an upper bound for the local hypomonotonicity required in Theorem 2.

Why does it matter to provide an underestimate of an upper bound?

How did you tune the parameters of the baselines for the experiments?

Did you use Q-values or counterfactual values for R-NaD and MMD for the experiments?

---

> ### Author Response · Authors · 2024-11-21
>
> Thank you for reviewing this paper and providing valuable comments. We have updated the manuscript according to the comments from all reviewers. Here, we reply to your questions and concerns about the paper.
>
> **[Weaknesses]**
>
> > POSGs can neither express untimeable games nor imperfect-recall games.
>
> Yes, we agree with that. We think timeability and perfect recall should be two standard assumptions for common equilibrium-learning algorithms in dynamic games. For clarity, we have specified the EFGs as perfect-recall EFGs.
>
> > Also, the reason they require a different representation is because treating them as NFGs is too expensive, not because they involve state.
>
> Yes, we agree with that. But treating them as NFGs actually results in an action space whose size is exponential in the number of distinct information states. The expense is somewhat related to states.
>
> > These sentences don't flow correctly.
>
> Thank you for pointing out the expression issues. We have revised them in our current submission.
>
> **[Questions]**
>
> > What's the catch here? Does this require exponential time for small quantal shocks?
>
> Like MMD [1], here we introduce the concept of quantal response equilibrium (QRE), which can be viewed as a class of regularized Nash equilibrium. We also borrow the NFG concept of Nash distribution to further establish a generalized concept in POSGs without relating to specific quantal shocks.
>
> > Is this claim based on experimental results for Kuhn poker? That would strike me as far from comprehensive enough to make such a sweeping claim.
>
> Thanks for the comment. We have replaced this claim, as it is indeed not suitable in Appendix D. However, the claim is based on several facts. First, hypomonocoticity relaxes the widely used monotonicity assumption [1, 2, 3] since monotonicity strictly requires $\lambda=0$ rather than $0\leq\lambda<\infty$. Theoretically, we prove that every NFG has a bounded $\lambda$ (Proposition 1). For dynamic games, we numerically estimate the order of magnitude for $\lambda$ to show that it should not be a large quantity in multiplayer Kuhn poker, which we subsequently use as a test environment.
>
> > Why does it matter to provide an underestimate of an upper bound?
>
> Thanks for the comment. We also find it inappropriate to specify the computed value as an underestimate and have removed this expression. Now we simply state that when the sampling is sufficient, the supremum of the computed value can be an estimate of a true $\lambda$ and thus reflect its order of magnitude.
>
> > How did you tune the parameters of the baselines for the experiments?
>
> For the step size $\eta$, we selected from $\\{0.001,0.002,0.005,0.01,0.1,1\\}$ for each algorithm. Since R-NaD and MMD do not provide experimental results in Kuhn poker with more than two players, we tuned their parameters based on the original settings for 2-player Kuhn poker when applying them to 3-player and 4-player scenarios. For the NFGs, we found the original parameter setting of moving-magnet MMD can already guarantee good performance and thus did not change it.
>
> > Did you use Q-values or counterfactual values for R-NaD and MMD for the experiments?
>
> For R-NaD, we used counterfactual values, which its original paper specifies. For MMD, we also used counterfactual values and verified the algorithm in 2-player Kuhn poker. However, we find that they no longer demonstrate a stable performance as DRDA when Kuhn poker has more than two players with more ranks, even if we have tried a range of parameters under different orders of magnitude.
>
> **References**:
>
> [1] Sokota S, D'Orazio R, Kolter J Z, et al. A unified approach to reinforcement learning, quantal response equilibria, and two-player zero-sum games. International Conference on Learning Representations, 2023.
>
> [2] Abe K, Ariu K, Sakamoto M, et al. Adaptively perturbed mirror descent for learning in games. International Conference on Machine Learning, 2024.
>
> [3] Perolat J, Munos R, Lespiau J B, et al. From Poincaré recurrence to convergence in imperfect information games: Finding equilibrium via regularization. International Conference on Machine Learning, 2021.
>
> Thanks again for your comments. We are looking forward to having further discussions with you.

---

> ### Comment · Reviewer_LUwx · 2024-11-27
> **Response**
>
> > Like MMD [1], here we introduce the concept of quantal response equilibrium (QRE), which can be viewed as a class of regularized Nash equilibrium. We also borrow the NFG concept of Nash distribution to further establish a generalized concept in POSGs without relating to specific quantal shocks.
>
> This doesn't isn't really what I was trying to get at. To put it more directly: computing a equilibria in general-sum games is hard from a complexity perspective. How does this jive with DRDA's guarantees? Will it be very slow in the worst case?

---

> > ### Author Response · Authors · 2024-11-27
> >
> > We are sorry for the misunderstanding and thank you for the clarification. Computing/approximating Nash equilibrium in general-sum games is a PPAD-hard problem. This complexity result describes the **worst-case** difficulty of solving general games, and it suggests that no existing algorithm can efficiently find NE in all games under no assumption. It does not contradict the fact that some games with specific properties (e.g., two-player zero-sum) can be solved within polynomial time. In this paper, the convergence of DRDA to Nash equilibrium also requires certain assumptions on game property. For single-round DRDA to directly converge to an exact NE rather than a QRE, it requires the game to be (strictly) monotone so that we can use an arbitrarily small $\epsilon$ (quantal shock). For multi-round DRDA to converge to NE, it requires the Nash equilibrium (corresponding to the rest point of DA) to be unique. Such assumptions make it possible for DRDA to approximate Nash equilibrium within polynomial time. In the worst-case games (with no specific restriction), there is no guarantee for DRDA to find a Nash equilibrium.

---

> > > ### Comment · Reviewer_LUwx · 2024-11-27
> > > **Response**
> > >
> > > Thanks for your responses. Some of my concerns were addressed, so I raised my score.

---

### Official Review · Reviewer_KwQ7 · 2024-11-03

**Soundness:** 3
**Presentation:** 3
**Contribution:** 3
**Rating:** 6
**Confidence:** 4

**Summary:**

This paper introduces Divergence-Regularized Discounted Aggregation (DRDA), a multi-round learning method designed for partially observable stochastic games. DRDA builds on a discounted version of Follow the Regularized Leader, adapting it for POSGs to address imperfect information and infinite horizons. The authors expand on previous work showing last-iterate convergence to quantal response equilibrium by defining a generalized Nash distribution, which extends the QRE concept to POSGs via divergence regularization.

**Strengths:**

1. This paper builds on prior work by extending last-iterate convergence from NFGs and EFGs to POSGs, with theoretical proofs, broadening the method's applicability.
2. It introduces the concept of a generalized Nash distribution, enhancing the QRE framework through divergence regularization.
3. Convergence proofs are provided for both single-round and multi-round DRDA, strengthening the method's theoretical foundation.
4. Experimental results demonstrate that DRDA outperforms existing methods across various benchmarks.

**Weaknesses:**

See Questions.

**Questions:**

1. The paper uses a more general POSG setting compared to traditional EFGs. What are the core challenges unique to this choice, and how do they affect algorithm design?
2. Multi-round DRDA resembles a variant of Perturb-based methods from previous work [1]. What advantages does DRDA offer over these established approaches?
3. Can prior methods like R-Nad [2] or APMD [1] be directly applied to POSGs, or are there specific limitations in these algorithms that DRDA addresses?

---

> ### Author Response · Authors · 2024-11-21
>
> Thank you for reviewing this paper and providing valuable comments. We have updated the manuscript according to the comments from all reviewers. Here, we reply to your questions and concerns about the paper.
>
> **[Questions]**
>
> _**Q1**_: The paper uses a more general POSG setting compared to traditional EFGs. What are the core challenges unique to this choice, and how do they affect algorithm design?
>
> The core challenge of using POSG in place of traditional EFG is that POSG can have an infinite horizon, which existing research on EFG hardly deals with. When the time horizon is infinite, the counterfactual value commonly used in the EFG algorithms (like R-NaD) becomes undefined, and we need to re-examine the convergence of learning dynamics under a general action value defined by Eq (3).
>
> _**Q2**_: Multi-round DRDA resembles a variant of perturb-based methods from previous work [1]. What advantages does DRDA offer over these established approaches?
>
> APMD is close to MMD with respect to the update formula and close to R-NaD and DRDA with respect to the learning scheme. The four methods all exhibit certain last-iterate convergence. The theoretical results of APMD and MMD can be directly applied to NFGs, and R-NaD focuses on perfect-recall EFGs under counterfactual value. In comparison, DRDA examines general infinite-horizon POSGs under action value. Besides, the last-iterate convergence of APMD, MMD, and R-NaD all require the monotonicity of the game, while DRDA only requires hypomonotonicity, which is a much less stringent assumption.
>
> _**Q3**_: Can prior methods like R-NaD [2] or APMD [1] be directly applied to POSGs, or are there specific limitations in these algorithms that DRDA addresses?
>
> In principle, R-NaD or APMD can be applied to POSGs by replacing the value function with a general action value. However, no existing theoretical or experimental result suggests that the algorithms can still converge under this condition. DRDA addresses this limitation since it is directly established in POSGs under action value. Besides, even using counterfactual values, R-NaD and MMD still cannot converge in multiplayer Kuhn poker (Section 5.2). This suggests that DRDA could be more suitable for solving multiplayer dynamic games.
>
> **References**:
>
> [1] Abe K, Ariu K, Sakamoto M, et al. Adaptively perturbed mirror descent for learning in games. International Conference on Machine Learning, 2024.
>
> [2] Perolat J, Munos R, Lespiau J B, et al. From Poincaré recurrence to convergence in imperfect information games: Finding equilibrium via regularization. International Conference on Machine Learning, 2021.
>
> Thanks again for your comments. We are looking forward to having further discussions with you.

---

> > ### Comment · Reviewer_KwQ7 · 2024-11-26
> >
> > Thank you for your response！

---

### Official Review · Reviewer_RHfE · 2024-11-03

**Soundness:** 3
**Presentation:** 3
**Contribution:** 3
**Rating:** 8
**Confidence:** 4

**Summary:**

This paper introduces a new algorithm for finding a Nash equilibrium in partially observable stochastic games (POSGs), which include setting with an infinite horizon.
The algorithm, abbreviated DRDA, is inspired by previous work on finding epsilon-regularized Nash equilibria, which have limitations of a finite horizon and may not always have convergence guarantees in a multi-player setting.
They generalize the epsilon-regularized Nash equilibria to include the discounting factor of infinite-horizon POSGs, to form what they call a generalized Nash distribution (GND).
DRDA can be formulated as an ODE, which is solved repeatedly in multiple rounds to converge to an NE in the limit.

While the theory for convergence requires an assumption that is not clear if it can be satisfied in all games (rest point of DRDA is unique => GND has bounded deviation from NE) and the theory of multi-round DRDA converge is not well established, experiments show that the DRDA converges to a high precision in small games, at a near-exponential rate.
Moreover, they provide linear convergence bounds under some hypomonotonicity assumptions.

**Strengths:**

- Paper is well written, given the high complexity of the topic.
- They show linear convergence if epsilon > lambda, the hypomonotonicity constant. I find the hypomonotonicity intriguing. I think it's likely to  see future follow-ups on this work.

**Weaknesses:**

- The theorems 1, 2 and 4 look similar to the results found in [Sokota] and [Perolat], extended to the POSGs. I did not investigate the exact differences, but these links deserve more highlights. Connection to MMD is written about in the Appendix, but as the update equations are written in two different notations I found it hard to follow in a limited time.
- Connected to previous the previous point, in experiments they compare with moving-magnet version of MMD, as L431 "MMD is close to DRDA", which they also abbreviate as MMD, while this version is substantially different from the non-moving-magnet base case. (Perhaps MMM would be a better abbreviation.) This makes me concerned that the real innovation of this paper is showing linear convergence under the hypomonotonicity assumption.
- I would enjoy more thorough discussion of the experiment results, see also the questions.
- Appendix E: The Euler method used for single-round DRDA -- SDRDA -- has a learning rate whose schedule is not mentioned in the paper and its relation to the rest point convergence in Theorem 2 is not clear. Also, in experiments they used MDRDA, which uses the Euler-based SDRDA. At first, based on the main paper text, I was under the impression that the discretization of Eq (6) is used for DRDA in the experiments, but the actual algorithm used is different. What is considered as an iteration exactly? One step of the Euler algorithm? It would be helpful to correctly label the algorithm in experiments as MDRDA.

**Questions:**

- How does multi-round DRDA depend on the gap of single-round DRDA from its rest point? If I understand correctly, this is unknown?
- How do you explain the "spikes" in Figure 1/2 at the beginning of each new round?
- In Figure 2, 3-player Kuhn, the "levels" (ignoring the spikes) are not monotonic for DRDA. I find it surprising. How is it possible it converges overall? Is there a measurable quantity that is monotonic? (perhaps regret? or gap from regularized eq?)
- Finding NE outside of constant-sum games is PPAD-hard as I'm sure the authors are well aware of. Perhaps it is hard to remove the uniqueness assumption because of that?
- Can you please run the algorithm on typical POSG games like the tiger, and compare to methods like HSVI?
- Can you please compare the algorithm also with Smoothed Predictive Regret Matching+ (with restarts)?
- How much tuning did the experiments require? The discussions is missing.

---

> ### Author Response · Authors · 2024-11-21
>
> Thank you for reviewing this paper and providing valuable comments. We have updated the manuscript according to the comments from all reviewers. Here, we reply to your questions and concerns about the paper.
>
> **[Weaknesses]**
>
> _**Weakness 1**_: Thanks for the comment. Currently, we have highlighted that we examine the convergence to a kind of QRE **under action value in general POSGs**. In comparison, MMD [Sokota] focuses on the original QRE concept in NFGs, and R-NaD [Perolat] considers a kind of QRE under counterfactual value in EFGs. For the connection to MMD, we have aligned the notations in Appendix E.2 of our current submission.
>
> _**Weakness 2**_: Thanks for the comment. In fact, the relationship between DRDA and MMD can be established **only when the base/magnet policy is fixed as a uniform policy**. Under general divergence regularization, the behavior of the two learning dynamics can be quite different, no matter whether the base/magnet policy is changing or not.
>
> _**Weakness 4**_: Thanks for the comment. For clarity, we have revised our experiment descriptions in Section 5 and Appendix E.1. For the $\alpha_m$ in the original submission, we clarify that it should be **a fixed step size $\eta$ rather than a scheduled learning rate** since we only use the simplest Euler's method. In our experiments, the rest-point convergence for single-round DRDA is guaranteed when $\eta$ is kept below the level of $10^{-2}$. Besides, we clarify that SDRDA is the discretization we employ and that an iteration is exactly one step of the Euler algorithm.
>
> **[Questions]**
>
> _**Q1**_: How does multi-round DRDA depend on the gap of single-round DRDA from its rest point? If I understand correctly, this is unknown?
>
> If we understand correctly, the $\mathcal{V}$ in Section 4.2 is the gap that you refer to? This gap approaches zero exponentially fast by Theorem 2 but can still exist in practice. However, the definition of multi-round DRDA requires an oracle $\mathcal{M}$ that assumes the gap to be zero. Therefore, **how this gap practically affects multi-round DRDA is indeed unknown**.
>
> _**Q2**_: How do you explain the "spikes" in Figure 1/2 at the beginning of each new round?
>
> We think **the spike is a result of switching the base policy in KL-divergence to the current policy**. Since the current policy is not an exact Nash equilibrium, it is unstable under the new regularization (using itself as the base). Therefore, at the beginning of the next round, it quickly comes to another region of policies, which may have a relatively high NashConv.
>
> _**Q3**_: In Figure 2, 3-player Kuhn, the "levels" (ignoring the spikes) are not monotonic for DRDA. I find it surprising. How is it possible it converges overall? Is there a measurable quantity that is monotonic? (perhaps regret? or gap from regularized eq?)
>
> Yes, this phenomenon suggests that it may be impractical to directly prove the monotonicity for the NashConv gap itself. However, **it is likely that an upper bound of NashConv at the rest point monotonically comes to zero**. The exact form of this upper bound, though unknown for now, could be the key to characterizing the convergence of multi-round DRDA. This quantity may not relate to regret, as last-iterate NashConv has no direct relationship with regret in multiplayer games. For example, OFTRL is no-regret, but its last iterate is away from Nash equilibrium in our 3-player NFG experiment. However, we think your second hypothesis deserves further investigation. This quantity could be the equilibrium gap in a game under certain divergence regularization, e.g., using an NE policy as the base. When the original game has a unique Nash equilibrium (corresponding to the unique DA rest point), this gap also approaches zero as the policy approaches the unique NE.
>
> _**Q4**_: Finding NE outside of constant-sum games is PPAD-hard as I'm sure the authors are well aware of. Perhaps it is hard to remove the uniqueness assumption because of that?
>
> Yes, we agree that **the uniqueness assumption serves to keep the problem out of PPAD-complete**. However, as is claimed in our response to Reviewer W41z (Q3), we find that **the uniqueness of rest point may not be a necessary condition** since it can be somewhat replaced by other distinct assumptions like individual concavity. It may also require an auxiliary assumption to establish an explicit convergence rate for multi-round DRDA.

---

> ### Author Response · Authors · 2024-11-21
>
> _**Q5**_: Can you please run the algorithm on typical POSG games like the tiger, and compare to methods like HSVI?
>
> Currently, we have run a 5-round DRDA ($10^3$ iterations per round, $\epsilon=0.1, \eta=0.01$) in **Adversarial Tiger** game and **Competitive Tiger** game under time horizon $H\in\\{2,3,4\\}$ and compared our result with the best result reported in [1]. Among HSVI, random search, informed search, SFLP, and CFR+ algorithms, SFLP demonstrates the best performance in all six cases (reported in Table 2 of [1]). However, **SFLP is clearly outperformed by DRDA** when we align the performance measure using NashConv:
>
> In Adversarial Tiger with $H=4$, SFLP is reported to achieve a 0.32 NashConv using 8 sec, while DRDA can achieve a 0.017 NashConv in 8 sec (terminating with a 0.010 NashConv in 10 sec).
>
> In Competitive Tiger with $H=3$, SFLP is reported to achieve a 0.36 NashConv using 48 sec, while DRDA achieves a 0.028 NashConv within 3 sec.
>
> In Competitive Tiger with $H=4$, SFLP is reported to achieve a 0.48 NashConv using 14 min, while DRDA achieves a 0.024 NashConv within 5 min.
>
> For the other three cases, SFLP is reported to achieve NashConv below 0.16 or 0.24 in 1 sec, while DRDA terminates with NashConv below $10^{-4}$ also in 1 sec. We have additionally provided the tiger game results in Section 5.2 (Figure 3) and Appendix F.4 (Table 7) of our current submission.
>
> _**Q6**_: Can you please compare the algorithm also with Smoothed Predictive Regret Matching+ (with restarts)?
>
> Yes, we have run stable/smooth PRM+ (with restart/projection; see [2]) in the NFGs and EFGs. However, we find the two techniques (stabilizing/smoothing) do not have an actual effect on the original PRM+ except in our 3-action bimatrix game, as the individual maximum regrets do not come below $1$ after initialization in the other 4 game scenarios. For the NFGs, the last iterates of PRM+ do not converge, while the stabilized one in the 3-action bimatrix game eventually converges at a high precision. In 3-player Kuhn poker, PCFR+ exhibits certain last-iterate convergence around $10^{-6}$. In 4-player Kuhn poker, PCFR+ oscillates around $10^{-3}$. In comparison, DRDA is eventually at the level of $10^{-10}$ and $10^{-6}$, respectively. Therefore, **DRDA has a clear advantage over PRM+ in 4 out of the 5 games**. We have additionally provided the last-iterate learning curves for PCFR+ in Section 5.2 of our current submission.
>
> _**Q7**_: How much tuning did the experiments require? The discussion is missing.
>
> For the step size $\eta$, we selected the best one from $\\{0.001,0.002,0.005,0.01,0.1,1\\}$ for each algorithm. The other parameters for DRDA and R-NaD were slightly tuned to guarantee the two algorithms at least converge in each single round. The parameters of moving-magnet MMD were tuned based on the setting mentioned in its original paper. In Appendix E.3 of our current submission, **we further provide a discussion about the parameter selection for DRDA**.
>
> **References**:
>
> [1] Delage A, Buffet O, Dibangoye J S, et al. HSVI can solve zero-sum partially observable stochastic games. Dynamic Games and Applications, 2024, 14(4): 751-805.
>
> [2] Farina G, Grand-Clément J, Kroer C, et al. Regret matching+: (In)stability and fast convergence in games. Advances in Neural Information Processing Systems, 2024, 36.
>
> Thanks again for your comments. We are looking forward to having further discussions with you.

---

> > ### Comment · Reviewer_RHfE · 2024-11-28
> >
> > Thank you very much for the answers and updated experiments! As they make the paper stronger, I am raising my rating.
> > I believe this paper will have interesting follow-ups and having such thorough experiments will help.

---

### Official Review · Reviewer_W41z · 2024-11-04

**Soundness:** 3
**Presentation:** 3
**Contribution:** 3
**Rating:** 6
**Confidence:** 3

**Summary:**

The paper introduces a new variant of multi-round discounted FTRL (called DRDA) which works in general (n-player, general-sum, partially observable, simultaneous or sequential action) games. The paper shows that the new algorithm converges (last-iterate) to a generalized Nash distribution (a Nash equilibrium of a regularized version of the game). Under an assumption (unique rest point), multi-round DRDA is proved to converge to an NE.

**Strengths:**

General games are hard. Any work that attempts to learn and prove convergence in this setting is important to the community.

Last-iterate convergence is valuable, especially compared to average-iterate algorithms that are common in the field.

Experimental results look impressive.

**Weaknesses:**

No discussion on the assumptions.

Tabular.

Single experiment seed runs.

**Questions:**

Q1: Can you say anything about what equilibria is being selected by DRDA compared to other methods?

Q2: How realistic is the local hypomonotonicity assumption?  I don’t think this is addressed in the text.

Q3: How realistic is the single rest point assumption? Is the single rest point a feature of the game, learning dynamics, or both? Do the games in the  experiments section have this property? What happens if they do not have this property?

Q4: The experiments look like single seed runs. Did you run a parameter sweep?

Minor:

* Line 16: “General value function”. I was unsure what this meant at first read through. A Non-zero sum value function?
* Line 39: Go is more naturally formulated as an EFG. Is there a more natural example for MGs?
* Line 69: If I am reading this right, “Nash distribution” is distinct from “Nash equilibrium”?
* Line 109: Is the set of all joint observations ever used? It seems like a strange quantity.
* Line 169/184/203/…: Please use inline math styling.
* Line 169: I am curious if using sum over player convs (rather than max) is important? I realize both would be zero at equilibrium.
* Line 183: Should a have a superscript: a^i? Because it is player i’s action, rather than a joint action?
* Line 225: Is this not more like a q-value? Why use v rather than q for the notation?
* Line 299: Does a GND always exist? It is not clear to me since any perturbation away from pi_base will result in: u^i(pi^i, pi^01_*) - u^i(pi_*) <= epsilon * (some negative value)

Score:

I am willing to raise the score after discussion with co-reviewers. Although I like the results, my main concern is how novel the contributions are compared to previous works. I would also like to be reassured that the method routinely converges -- I do not have a good intuition of how strong of an assumption single-resting point is. Furthermore, any answers to the questions I have asked will also likely improve my understanding and appreciation of the work.

---

> ### Author Response · Authors · 2024-11-21
>
> Thank you for reviewing this paper and providing valuable comments. We have updated the manuscript according to the comments from all reviewers. Here, we reply to your questions and concerns about the paper.
>
> **[Questions]**
>
> _**Q1**_: Can you say anything about what equilibria is being selected by DRDA compared to other methods?
>
> Like magnetic mirror descent (MMD) and regularized Nash dynamics (R-NaD), single-round DRDA finds a kind of quantal response equilibrium (QRE). However, DRDA examines the convergence to a type of QRE **under action value in general POSGs**. In comparison, MMD focuses on the original QRE concept in NFGs, and R-NaD studies a type of QRE under counterfactual value in EFGs. For DRDA with multiple rounds, it aims to find the Nash equilibrium in a less and less perturbed game (which eventually becomes the original game).
>
> _**Q2**_: How realistic is the local hypomonotonicity assumption? I don’t think this is addressed in the text.
>
> Hypomonotonicity relaxes monotonicity by allowing $\lambda$ to be non-zero, and local hypomonotinicity further relaxes (global) hypomonotonicity by allowing the property to hold in a local policy set rather than the entire $\Pi$. In the paper, we have actually provided some evidence to show that global hypomonotonicity is already a realistic assumption. For static games, we theoretically prove every NFG has a bounded $\lambda$ (Proposition 1). For dynamic games like Kuhn poker, we estimate the order of $\lambda$ in Appendix D and find it should not be a large quantity.
>
> _**Q3**_: How realistic is the single rest point assumption? Is the single rest point a feature of the game, learning dynamics, or both? Do the games in the experiments section have this property? What happens if they do not have this property?
>
> The uniqueness of rest point in DRDA is analogous to the uniqueness of equilibrium in a game (regularized under corresponding $\epsilon$ and $\pi_{base}$). Since the learning dynamic is actually induced by the game, the uniqueness of rest point is essentially a game feature. Some existing theoretical works on last-iterate convergence (e.g., [1] and [2]) also assume the uniqueness of NE, which is not guaranteed to hold in real-world games. However, even when the uniqueness of the rest point does not hold, DRDA can still work in reality. For example, in Appendix B.2 of our current submission, we prove the same NashConv bound in addition to Theorem 3 when the game is individually concave (e.g., the NFGs in the experiment). That is to say, the uniqueness of the rest point should not be viewed as a necessary condition.
>
> _**Q4**_: The experiments look like single seed runs. Did you run a parameter sweep?
>
> Yes, the algorithms do not involve randomness when starting from a uniform policy (by default). However, we did run the algorithms under different parameter settings. For the step size $\eta$, we selected the best one from $\\{0.001,0.002,0.005,0.01,0.1,1\\}$ for each algorithm. The other parameters for DRDA and R-NaD were slightly tuned to guarantee the two algorithms at least converge in each single round. The parameters of moving-magnet MMD were tuned based on the setting mentioned in its original paper. In Appendix E.3 of our current submission, we further provide a discussion about the parameter selection for DRDA.
>
> **[Minor]**
>
> Line 16: We say the (action) value function is "general" because it considers both imperfect information and infinite horizon. In comparison, the widely used concept of counterfactual value is undefined when the time horizon is infinite.
>
> Line 39: Yes, pursuit-evasion games and video games like fighting games are naturally formulated as MGs.
>
> Line 69: Yes, Nash distribution is a type of QRE and is distinct from Nash equilibrium. It can be viewed as an equilibrium in a perturbed/regularized game, while NE is for the original game.
>
> Line 109: This set is not explicitly used in this paper, but it is generally required in the POSG definitions.
>
> Line 169/184/203/…: Thank you for mentioning the styling. We have revised them in our current submission.
>
> Line 169: No, it is not important. What you refer to could be directly called exploitability and is also commonly used.
>
> Line 183: The notation $a$ here is different from \vec{a} and actually indicates $a^i$. Since all players share the same action set $\mathcal{A}$, we do not specify the superscript of $a$ here in order to match the subscript of $\pi^i_a$.
>
> Line 225: Yes, it is actually a state-action value. But it is just common to use a general notation $v$ as such (not related to the function $V(\cdot)$) when it comes to defining the (hypo)monotonicity of the game.

---

> > ### Comment · Reviewer_W41z · 2024-11-25
> >
> > Thank you for the clarifications. I have no further questions.

---

> ### Author Response · Authors · 2024-11-21
>
> Line 299: Currently, we also find it hard to directly prove that it always exists. In our current submission, we add the existence of GND as an extra condition in the second statement of Theorem 3. Still, the assumptions in Theorem 4 and the first statement of Theorem 3 do not change since Nash equilibrium (i.e., GND under $\epsilon=0$) always exists. For the definition of GND, it serves to relate a game to a learning dynamic under the same regularization. In NFGs, Nash distribution corresponds to the rest point of discounted FTRL. For POSGs, Definition 4 provides a GND concept that can "characterize" the rest point of DRDA (by Theorem 1). Actually, after examining a range of potential GND definitions, we find that Lemma 3 and Theorem 1 still hold if we replace the terms $\Pr\left(h _ t|\pi _ *\right)$ in Definition 4 with $\Pr\left(h _ t|(\pi ^ i,\pi _ * ^{-i})\right)$. This provides another form of GND definition, and we only require a policy to satisfy either one when applying Theorem 3 in practice.
>
> **References**:
>
> [1] Daskalakis C, Panageas I. Last-iterate convergence: Zero-sum games and constrained min-max optimization. 10th Innovations in Theoretical Computer Science, 2019.
>
> [2] Lee C W, Kroer C, Luo H. Last-iterate convergence in extensive-form games. Advances in Neural Information Processing Systems, 2021, 34: 14293-14305.
>
> Thanks again for your comments. We are looking forward to having further discussions with you.

---

### Author Response · Authors · 2024-11-27
**Response to All Reviewers**

Dear reviewers,

Thank you for reviewing this paper. We greatly appreciate your effort in reading the paper and providing valuable comments. Your suggestions are quite helpful for us to improve the quality of this paper. Based on your comments and suggestions, we have made a revision to our manuscript and updated the submission. The major modifications involve Section 1 (Introduction), Section 5 (Experiments), Appendix B.2 (Related to Individual Concavity), Appendix D (Game Hypomonotonicity), Appendix E (Implementation Details), and Appendix F.4 (Tiger Game). The additional/revised contents have been highlighted in blue/green. We will be happy if our responses and modifications can address your concerns and improve your appreciation of this work. If you find something unclear, please feel free to point it out. We are more than delighted to have further discussions and improve our manuscript.

---

### Meta-Review · Area_Chair_D9T3 · 2024-12-10

**Metareview:**

This paper studies equilibrium-computation in partially observable stochastic games (POSGs), by developing a Divergence-Regularized Discounted Aggregation approach. Focusing on the continuous-time dynamics, last-iterate convergence was established, locally and asymptotically. Overall, the paper is well-written and easy to follow. Despite the mostly positive reviews, after more careful reading myself, I have some reservations about recommending a clear acceptance.

First, the idea of adding (magnet) regularization to obtain last-iterate convergence in learning in games is not that novel, see e.g., earlier works [1,2,3,4,6,7] (in addition to what has been mentioned in the paper). In particular, for EFGs, [1] showed that last-iterate to QRE can be obtained without unique NE assumption, and the algorithm there is discrete-time, and the convergence results are non-asymptotic and global; for MGs (including NFGs), [2,3] introduced the idea of regularization and obtained last-iterate convergence that is "non-asymptotic" and "global" also; for "discounted FTRL" (as named in the paper, which is essentially a continuous-time version of the update rule in [5], as also pointed out by the authors), [4] has global (asymptotic) convergence guarantees, exactly based on [5], for Markov games; the "anchor-policy" regularized idea, which is essentially the single-round DRDA, in fact first appeared and validated in [6,7] (to my knowledge). However, these closely-related results (some with stronger guarantees) were not mentioned nor compared numerically, making it a bit hard and misleading for the readers/reviewers to accurately assess the contributions of the paper. The existence of MMD [8] further weakened the novelty of the proposed dynamics, as they share quite some similarities (as also pointed out in the Appendix). It would have been better if these results were well acknowledged in the main paper, and the new dynamics were sold as "extensions" or at least "inspired" by these algorithms.

Second, it would have been better if the theoretical contribution was stronger. The results were not only asymptotic and local, but also relied on strong assumptions such as the existence of a (local) compact region such that $\lambda$-hypomonotonicity is satisfied and contains a rest point, the initialization within such a region, as well as the choice of $\epsilon$ that depends on $\lambda$; the uniqueness of DA's rest point, etc. In particular, the results in Theorem 4 for multi-round DRDA relied on "the convergence of the rest-point sequences" (together with the uniqueness of the rest-point), which IMO should be part of the "convergence guarantees" one needs to prove. The only case where the $\lambda$-hypomonotonicity is "global" is the NFG case (for which there already exist stronger results), which made the significance of the condition a bit weaker, especially given that the title and the theme of the paper is to "unify" results under the POSG framework (beyond NFGs). It was not very clear what the technical challenges were in proving the main theorems (compared to the existing results mentioned above, and given some key results from [9]). It would have been a stronger case if the experimental results were more extensive. However, compared to other closely-related works, e.g., [6,7,8], the experiments mostly focused on small-scale problems, and it was not clear if the methods scale well in larger games, as [6,7,8].

Third, the technical rigor and clarity of the exposition may need a bit more care. It is known that POSGs are very hard to solve in general, not to mention "general-sum" POSGs. The current title, abstract, and writing in several places left the impression (especially for the general audience) that the proposed algorithms "solve" POSGs, which is not the case given the above comments on the theoretical results. Moreover, it was emphasized that the authors considered POSGs in order to "unify" NFGs, EFGs, and MGs. However, it is not clear any other POSGs beyond these cases can be covered. In fact, in the main results, it is not clear whether any of these games, or in fact, any POSGs, satisfy the "(local) $\lambda$-hypomonotonicity" property, besides the NFG. Also, as Reviewer LUwx pointed out, EFGs without perfect recall cannot be covered by POSGs, and, although general POSGs (without the tree structure) can still be represented by EFGs, the representation blows up. It would have been better to only claim contributions to NFG, EFG with perfect recall, and MGs (instead of general POSGs). But still, connecting the assumptions in the theory with the structures of these games would help make the case.

Overall, I found this paper a borderline case, and suggest the authors carefully incorporate the feedback from this round (from both the reviewers and the area chair) in preparing the next version of the paper.

**Additional Comments On Reviewer Discussion:**

There were some concerns regarding the rigor and clarity of the exposition. In particular, Reviewer LUwx pointed out that POSGs cannot cover EFGs without perfect recall, and the authors acknowledged and corrected in the revision. There were also some concerns regarding the (over-)claims in the paper, and experimental details. The authors addressed most of such comments. However, some answers were not very rigorous (to me), e.g., it was claimed in the rebuttal to Reviewer LUwx that "Such assumptions make it possible for DRDA to approximate Nash equilibrium within polynomial time". However, given the results were local, and the dependence on the problem-dependent parameters was unclear, such an argument seems not true (even under the assumptions the authors claimed). Overall, I appreciate the authors' rebuttal and revision, but also feel like more care may be needed to make it a stronger paper.

[1] Liu, M., Ozdaglar, A. E., Yu, T., & Zhang, K. The Power of Regularization in Solving Extensive-Form Games. In The Eleventh International Conference on Learning Representations.

[2] Cen, S., Wei, Y., & Chi, Y. (2021). Fast policy extragradient methods for competitive games with entropy regularization. Advances in Neural Information Processing Systems, 34, 27952-27964.

[3] Cen, S., Chi, Y., Du, S. S., & Xiao, L. Faster Last-iterate Convergence of Policy Optimization in Zero-Sum Markov Games. In The Eleventh International Conference on Learning Representations.

[4] Sayin, M., Zhang, K., Leslie, D., Basar, T., & Ozdaglar, A. (2021). Decentralized Q-learning in zero-sum Markov games. Advances in Neural Information Processing Systems, 34, 18320-18334.

[5] Leslie, D. S., & Collins, E. J. (2005). Individual Q-learning in normal form games. SIAM Journal on Control and Optimization, 44(2), 495-514.

[6] Athul Paul Jacob, David J Wu, Gabriele Farina, Adam Lerer, Hengyuan Hu, Anton Bakhtin, Jacob Andreas, and Noam Brown. Modeling strong and human-like gameplay with kl-regularized search. In International Conference on Machine Learning, pp. 9695–9728. PMLR, 2022.

[7] Bakhtin, A., Wu, D. J., Lerer, A., Gray, J., Jacob, A. P., Farina, G., ... & Brown, N. Mastering the Game of No-Press Diplomacy via Human-Regularized Reinforcement Learning and Planning. In The Eleventh International Conference on Learning Representations.

[8] Samuel Sokota, Ryan D’Orazio, J Zico Kolter, Nicolas Loizou, Marc Lanctot, Ioannis Mitliagkas, Noam Brown, and Christian Kroer. A unified approach to reinforcement learning, quantal response equilibria, and two-player zero-sum games. In International Conference on Learning Representations, 2023.

[9] Panayotis Mertikopoulos and Zhengyuan Zhou. Learning in games with continuous action sets and unknown payoff functions. Mathematical Programming, 173:465–507, 2019.

---

### Decision · Program_Chairs · 2025-01-22

Accept (Poster)